# Vangl2 suppresses NF-κB signaling and ameliorates sepsis by targeting p65 for NDP52-mediated autophagic degradation

Jiansen Lu[1,2†], Jiahuan Zhang[3†], Huaji Jiang[2,4†], Zhiqiang Hu[2†], Yufen Zhang[2], Lian He[5,6], Jianwu Yang[2], Yingchao Xie[2], Dan Wu[2], Hongyu Li[2], Ke Zeng[2], Peng Tan[5,7], Qingyue Xiao[8], Zijing Song[8], Chenglong Pan[1], Xiaochun Bai[8]*, Xiao Yu[2,3,9]*

[1]Department of Joint Surgery, the Fifth Affiliated Hospital, Southern Medical University, Guangzhou, China; [2]Department of Immunology, School of Basic Medical Sciences, Southern Medical University, Guangzhou, China; [3]Department of Clinical Laboratory Medicine, Guangdong Provincial People's Hospital (Guangdong Academy of Medical Sciences), Southern Medical University, Guangzhou, China; [4]Department of Orthopaedics, Yuebei People's Hospital Affiliated to Medical College of Shantou University, Shaoguan, China; [5]Department of Pharmacology, School of Medicine, Southern University of Science and Technology, Shenzhen, China; [6]Institute of Biosciences and Technology, College of Medicine, Texas A&M University, Houston, United States; [7]Klarman Cell Observatory, Broad Institute of MIT and Harvard, Cambridge, United States; [8]Department of Cell Biology, School of Basic Medical Sciences, Southern Medical University, Guangzhou, China; [9]Guangdong Provincial Key Lab of Single Cell Technology and Application, Southern Medical University, Guangzhou, China

*For correspondence:
baixc15@smu.edu.cn (XB);
xiaoyu523@smu.edu.cn (XY)

†These authors contributed equally to this work

**Abstract** Van Gogh-like 2 (Vangl2), a core planar cell polarity component, plays an important role in polarized cellular and tissue morphology induction, growth development, and cancer. However, its role in regulating inflammatory responses remains elusive. Here, we report that Vangl2 is upregulated in patients with sepsis and identify Vangl2 as a negative regulator of The nuclear factor-kappaB (NF-κB) signaling by regulating the protein stability and activation of the core transcription component p65. Mice with myeloid-specific deletion of Vangl2 (Vangl2ΔM) are hypersusceptible to lipopolysaccharide (LPS)-induced septic shock. Vangl2-deficient myeloid cells exhibit enhanced phosphorylation and expression of p65, therefore, promoting the secretion of proinflammatory cytokines after LPS stimulation. Mechanistically, NF-κB signaling-induced-Vangl2 recruits E3 ubiquitin ligase PDLIM2 to catalyze K63-linked ubiquitination on p65, which serves as a recognition signal for cargo receptor NDP52-mediated selective autophagic degradation. Taken together, these findings demonstrate Vangl2 as a suppressor of NF-κB-mediated inflammation and provide insights into the crosstalk between autophagy and inflammatory diseases.

## eLife assessment

This **valuable** manuscript describes a novel role of Vangl2, a core planar cell polarity protein, in linking the NF-kB pathway to selective autophagic protein degradation in myeloid cells. The mechanistic studies provide **convincing** evidence that Vangl2 targets p65 for NDP52-mediated autophagic degradation, limiting inflammatory NF-kB response, with functional significance of the proposed mechanism in sepsis. Additional future studies dissecting autophagic Vangl2 functions in various

myeloid subsets in the context of inflammation could be informative, and additional Vangl2 targets in the inflammatory pathway, including IKK2, could also be explored. Overall, this exciting study can advance our understanding of NF-kB control, particularly in the context of inflammatory diseases.

## Introduction

Van Gogh-like 2 (Vangl2), a core planar cell polarity (PCP) component, mediates Wingless-type (Wnt)/PCP signaling, and controls homeostasis, development, and repair of organs (*Bailly et al., 2018*; *Brunt et al., 2021*; *Hatakeyama et al., 2014*). Vangl2 has four transmembrane domains with both carboxyl termini and amino oriented toward the cytoplasm, which is phosphorylated in the endoplasmic reticulum (ER) and then transported to the cell surface and becomes stabilized, while the unphosphorylated Vangl2 is unstable and internalized to degrade via the lysosomal pathway (*Feng et al., 2021*). The shuttle of Vangl2 between cytoplasm and cell membrane results in its multifunction, including adhesion, membrane protrusive activity, migration, and bridging proteins (*Hatakeyama et al., 2014*). Indeed, Vangl2 inhibited matrix metalloproteinase 2 (MMP2) activity and affected cell adhesion to extracellular matrix proteins (*Jessen and Jessen, 2017*). Vangl2 also modulates glomerular injury by promoting MMP9 (*Papakrivopoulou et al., 2018*). Moreover, the abnormal function of Vangl2 results in various diseases, such as cancer, kidney glomerular injury, idiopathic pulmonary fibrosis, and systemic dysplasia (*Papakrivopoulou et al., 2018*; *Poobalasingam et al., 2017*). Vangl2 is markedly downregulated in patients with emphysema (*Poobalasingam et al., 2017*), while which level was upregulated and amplified in breast, ovarian, and uterine carcinomas (*Kandoth et al., 2013*; *Cerami et al., 2012*; *Gao et al., 2013*). In addition, Vangl2 is bounded to p62 to promote breast cancer (*Puvirajesinghe et al., 2016*). Our previous study showed that Vangl2 prevents osteogenic differentiation in mesenchymal stem cells, resulting in osteogenic dysplasia (*Gong et al., 2021*). Vangl2-mediated downstream factors of Toll-like (TLR) or interleukin (IL)-1 receptor, such as myeloid differentiation factor 88 (MyD88) (*Gong et al., 2021*), suggesting that Vangl2 may play roles in immune-related diseases, including autoimmune diseases. However, the function of Vangl2 in inflammatory diseases remains uncovered.

The NF-κB signaling is critical for the pathogenesis of a number of inflammatory diseases, such as inflammatory bowel disease, rheumatoid arthritis, systemic lupus erythematosus, and sepsis (*Liu et al., 2017*). NF-κB activation relies on the pattern recognition receptors recognition of pathogen-associated molecular patterns, including TLR, nucleotide-binding oligomerization domain (NOD)-like receptors, and retinoic acid-inducible gene I (RIG-I)-like receptors (*Liu et al., 2017*). Lipopolysaccharide (LPS) triggers TLR activation and the recruitment of adaptor proteins including MyD88. This in turn activates a series of downstream canonical NF-κB signaling cascade, resulting in the phosphorylation and degradation of IκB and the nuclear translocation of RelA/p65 and p50 to induce the transcription of several inflammatory cytokines, such as IL-1, IL-6, and tumor necrosis factor-α (TNF-α) (*Funes et al., 2018*; *Locati et al., 2020*). Since uncontrolled immune responses are detrimental to the host, inappropriate or excessive NF-κB activity contributes to the pathogenesis of various inflammatory diseases and cancer (*Cartwright et al., 2016*). Thus NF-κB signaling must be tightly regulated to maintain immune balance in the organism. In recent decades, extensive studies have focused on the mechanisms underlying the regulation of NF-κB signaling. Recent research demonstrated that NLRC5 strongly prevents NF-κB signaling pathway by interacting with IκB kinase (IKK) α/IKKβ and blocking their phosphorylation (*Cui et al., 2010*). COMMD1, PPARγ, SOCS1, and GCN5 were also shown to negatively regulate NF-κB signaling (*Bartuzi et al., 2013*; *Mao et al., 2009*). Meanwhile, tripartite motif-containing protein 21 (Trim21) enhanced the interaction of p65 with IKK, which promotes p65 phosphorylation and downstream gene activation (*Yang et al., 2021*). However, the molecular mechanisms underpinning the regulation of NF-κB signaling are still elusive.

Autophagy, a conserved intracellular degradation pathway, decomposes cytoplasmic organelles and components, and acts as a defense mechanism to pathogen infection, playing a crucial role in nutrient recycling, stress response, and cellular homeostasis (*Ashrafi and Schwarz, 2013*; *Denton and Kumar, 2019*). Recent evidence indicates that autophagy is highly selective when delivering specific substrates to autolysosomal degradation by virtue of a number of cargo receptors, including sequestosome 1 (SQSTM1/p62), optineurin (OPTN), nuclear dot protein 52 (NDP52/CALCOCO2), and neighbor of BRCA1 (NBR1) (*Gatica et al., 2018*). Selective autophagy targets immune regulators

for degradation, thus suppressing innate immune signaling, such as type I IFN and NF-κB signaling (*Pradel et al., 2020*; *Tong et al., 2012*). Moreover, p62 protein has been identified as a novel Vangl2-binding partner (*Puvirajesinghe et al., 2016*), and our recent study has demonstrated that Vangl2 reduces lysosomal chaperone-mediated autophagy (CMA) activity by targeting LAMP-2A for degradation (*Gong et al., 2021*). However, whether and how Vangle2 is involved in the selective autophagic regulation of NF-κB signaling remains largely unknown.

In this study, we uncover a previously unrecognized role of Vangl2, as a 'molecular brake', in the negative regulation of NF-κB signaling to prevent excessive and potentially harmful immune responses during sepsis in both human patient samples and LPS-induced mouse model. Induction of Vangl2 upon inflammation recruits an E3 ubiquitin ligase PDLIM2 to catalyze K63-linked ubiquitination on p65, thus promoting the recognition of p65 by the cargo receptor NDP52 and resulting in the selective autophagic degradation of p65. Our findings provide a potential target for the treatment of inflammatory diseases.

## Results

### Loss of Vangl2 promotes inflammatory responses in LPS-induced septic shock

To investigate the potential role of Vangl2 in inflammatory response, we first analyzed the expression of Vangl2 in peripheral blood mononuclear cells (PBMCs) from sepsis patients (*Figure 1—figure supplement 1A*) and found mRNA level of Vangl2 was increased in the sepsis patients compared to healthy control (*Figure 1A*). Moreover, the serum of the high Vangl2 expression group had lower levels of IL-6, white blood cell, and acute C-reactive protein than the low Vangl2 expression group (*Figure 1—figure supplement 1A*). Expression of Vangl2 from database GSE156382 also confirmed that Vangl2 mRNA was induced during sepsis (*Figure 1—figure supplement 1B*). To further determine whether *Vangl2* expression could be regulated in response to inflammatory stimulation, we treated mice with LPS to activate NF-κB pathway and detected *Vangl2* mRNA in different tissues from LPS-treated mice, and found the expression of *Vangl2* significantly increased in secondary lymphoid organs including the spleen and lymph nodes after LPS stimulation, but not in other tissues (*Figure 1B*). Moreover, quantitative polymerase chain reaction (qPCR) and western blot analyses revealed strong upregulation of *Vangl2* at both the mRNA and protein levels after LPS stimulation in bone marrow (BM)-derived macrophages (BMDMs), neutrophils, and peritoneal macrophages (pMACs) (*Figure 1—figure supplement 1C–1E*), suggesting that Vangl2 is induced in immune organ tissues and cells in response to inflammation.

To determine the function of Vangl2 during LPS-induced sepsis, we specifically ablated Vangl2 in myeloid cells by crossing *Vangl2*^flox/flox^ mice with mice expressing lysozyme proximal promoter (*Lyz2-Cre*). The resultant homozygous *Vangl2*^flox/flox^ × *Lyz2*-Cre mice were designated *Vangl2*^ΔM^ mice and selective deletion of Vangl2 in myeloid cells of *Vangl2*^ΔM^ mice were confirmed by PCR (*Figure 1—figure supplement 1F*). Although there were no significant differences in the size of spleen and lymph nodes between wild-type (WT) and *Vangl2*^ΔM^ mice (*Figure 1—figure supplement 1G*), myeloid-specific loss of Vangl2 increased the number of monocytes, macrophages, and neutrophils in the spleen and BM (*Figure 1—figure supplement 1H–I*). To gain further insight into the physiological function of Vangl2, we treated WT and *Vangl2*^ΔM^ mice with a high dose of LPS and monitored mice survival. We found that all *Vangl2*^ΔM^ mice died after LPS-induced septic shock within 20 hr, compared to only 20% of the WT mice, whereas the remaining WT mice survived for more than 50 hr (*Figure 1C*). Consistent with this observation, *Vangl2*^ΔM^ mice markedly increased the protein level of IL-1β in isolated CD11b^+^ splenocytes (*Figure 1D*), and mRNA levels of *Il1b*, *Tnfa* and *Il6* in the spleen (*Figure 1E*). Meanwhile, *Vangl2*^ΔM^ mice showed markedly elevated serum amounts of proinflammatory cytokines such as IL-1β, TNF-α, and IL-6 after LPS treatment, compared with WT mice (*Figure 1F*). Together, these data provide in vivo evidence that myeloid cell-specific deletion of Vangl2 in mice enhances the sensitivity and severity of LPS-induced septic shock and is associated with increased expression of proinflammatory cytokines.

### Vangl2 negatively regulates NF-κB activation and inflammation in myeloid cells

To investigate the mechanisms by which Vangl2 prevents sepsis, we performed RNA-seq analysis to identify signal pathways involved in LPS-induced septic shock by comparing LPS-stimulated BMDMs

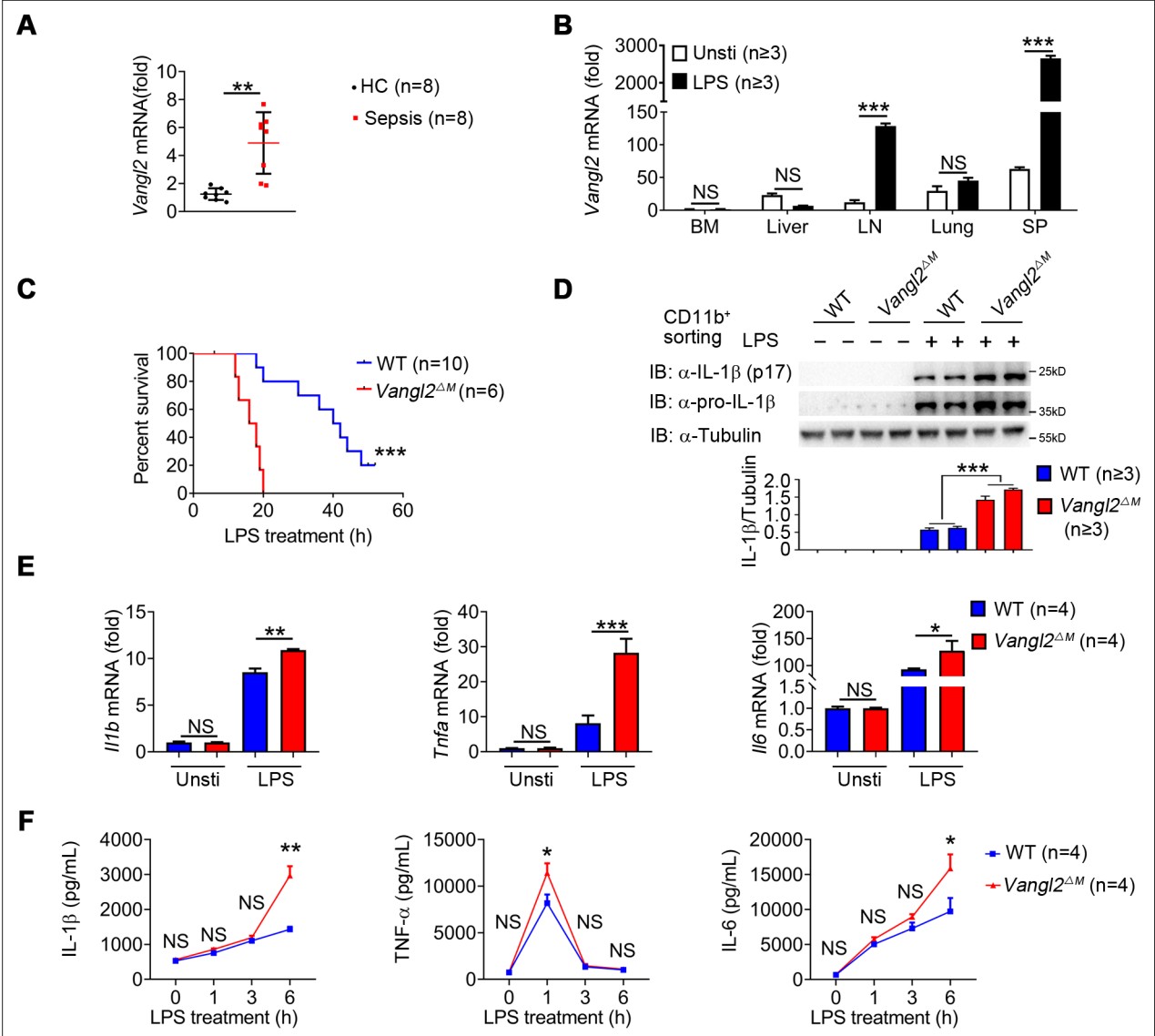

**Figure 1.** Van Gogh-like 2 (Vangl2) ablation promotes inflammation during LPS treatment. (**A**) Transcription levels of *Vangl2* in PBMCs from healthy volunteers (healthy control, HC) and sepsis patients were analyzed by real-time PCR (*n* = 8). (**B**) *Vangl2* mRNA in different organs from mice treated with or without LPS (*n* ≥ 3). (**C**) The survival rates of wild-type (WT) and *Vangl2*ΔM mice treated with high dosage of LPS (30 mg/kg, intraperitoneally [i.p.]) (*n* ≥ 3). (**D–F**) WT and *Vangl2*ΔM mice (*n* ≥ 3) were treated with LPS (30 mg/kg, i.p.). Splenocytes were collected at 9 hr after LPS treatment. Cell lysates of CD11b+ sorted splenocytes were analyzed by immunoblotting with the indicated antibodies (*n* ≥ 3) (**D**). RNAs from splenocytes were isolated and used for expression analysis of *Il1b*, *Tnfa*, and *Il6* using qPCR (*n* ≥ 3) (**E**). Sera were collected at indicated times post LPS treatment and subjected to enzyme-linked immunosorbent assay (ELISA) analysis of IL-1β, tumor necrosis factor-α (TNF-α), and IL-6 (*n* ≥ 3) (**F**). PBMCs, peripheral blood mononuclear cells; Unsti, unstimulation; LPS, lipopolysaccharide; LN, lymph node; SP, spleen. Data are representative of three independent experiments and are plotted as the mean ± standard deviation (SD). Two-tailed Student's t test for A. Multiple t tests for B, E and F. Log-rank (Mantel-Cox) test for survival curve. *p < 0.05, **p < 0.01, ***p < 0.001 vs. corresponding control.

The online version of this article includes the following source data and figure supplement(s) for figure 1:

**Source data 1.** Uncropped and labelled gels for *Figure 1*.

**Source data 2.** Raw data for *Figure 1*.

**Source data 3.** Raw unedited gels for *Figure 1*.

**Figure supplement 1.** Expression of Van Gogh-like 2 (Vangl2) during sepsis and lipopolysaccharide (LPS) treatment.

**Figure supplement 1—source data 1.** Uncropped and labelled gels for (*Figure 1—figure supplement 1*).

**Figure supplement 1—source data 2.** Raw data for (*Figure 1—figure supplement 1*).

**Figure supplement 1—source data 3.** Raw unedited gels for (*Figure 1—figure supplement 1*).

from *Vangl2*ᴬᴹ and WT mice. Results showed 907 upregulated genes (including inflammatory cytokines) and 1092 downregulated genes in response to Vangl2 deficiency in BMDMs after LPS stimulation (*Figure 2—figure supplement 1A*). Consistent with *Figure 1*, gene ontology analysis further identified that these genes are involved in cellular immune responses, including 'cellular response to lipopolysaccharide' (*Figure 2—figure supplement 1B*). Moreover, Kyoto Encyclopedia of Genes and Genomes (KEGG) analysis revealed genes involved in TNF signaling pathway and cytokine–cytokine receptor interaction were highly enriched in *Vangl2*ᴬᴹ BMDMs after LPS stimulation, suggesting that Vangl2 may regulate these signaling pathways and related cytokines release (*Figure 2—figure supplement 1C*).

To determine the function of Vangl2 in innate immune signaling in myeloid cells, we isolated BMDMs, neutrophils, and pMACs from WT and *Vangl2*ᴬᴹ mice, treated them with LPS, and performed immunoblot analysis with specific antibodies. We found that phosphorylations of p65 were enhanced and remained high level for a sustained period in Vangl2-deficient pMACs and neutrophils, compared with WT cells (*Figure 2A–D*). Consistent with this observation, Vangl2-deficient macrophages, and neutrophils showed markedly elevated proinflammatory cytokines such as TNF-α, IL-6 (*Figure 2E, F* and *Figure 2—figure supplement 1D, E*), and IL-1β (*Figure 2—figure supplement 1F*), after LPS treatment. Moreover, we detected enhanced p65 nuclear accumulation in LPS-induced Vangl2-deficient neutrophils, compared with WT neutrophils (*Figure 2G, H*).

To further confirm the function of Vangl2 in regulating NF-κB signaling, we overexpressed Flag-tagged Vangl2 in A549 cells. We found that Vangl2 overexpression inhibited phosphorylation of p65 in A549 cells (*Figure 2I, J*). These results suggest that Vangl2 prevents LPS-induced NF-κB activation and proinflammatory cytokine production.

## Vangl2 inhibits LPS-induced NF-κB activation by interacting with p65

To clarify the regulatory mechanism of Vangl2, we transfected Chinese hamster ovary (CHO) or 293T cells with NF-κB luciferase reporter vector, with or without the Vangl2 plasmid, then treated the cells with LPS, IL-1β, or TNF-α. We found that Vangl2 markedly inhibited NF-κB activation induced by LPS, IL-1β, or TNF-α in a dose-dependent manner (*Figure 3A–C*). Next, we sought to determine potential signaling molecules that mediated the NF-κB-luc reporter. NF-κB-luc activity was strongly activated by overexpression of MyD88, IRAK1, TRAF6, IKKα, IKKβ, or p65, but all of these activities were inhibited when *Vangl2* was co-transfected at increasing concentrations (*Figure 3D* and *Figure 3—figure supplement 1A–F*), suggesting that Vangl2 may block NF-κB activation at the very downstream signaling level of p65.

To test this prediction, we transfected 293T cells with HA-tagged Vangl2 together with Flag-tagged IKKα, IKKβ, p65, TRAF6, IRAK1, or MyD88. Co-immunoprecipitation (co-IP) assay revealed that Vangl2 interacted strongly with IKKα, IKKβ, p65, and MyD88 (*Figure 3E* and *Figure 3—figure supplement 1G, H*). In addition, endogenous co-IP immunoblot analyses showed that Vangl2 was strongly associated with p65 upon LPS stimulation in BMDMs, but not with IKKα, IKKβ, or MyD88 (*Figure 3F* and *Figure 3—figure supplement 1I*). Moreover, ZDOCK server predicted that Vangl2 may potentially interact with p65 by a hydrogen bond (*Figure 3—figure supplement 1J*). We further investigated the co-localization of Vangl2 and p65 by confocal microscopy and found a weak-co-localization of Vangl2 with p65 in unstimulated cells and the co-localization between Vangl2 and p65 was notably enhanced upon LPS stimulation (*Figure 3G* and *Figure 3—figure supplement 1K*). To determine how Vangl2 interacts with cytoplasmic p65, we isolated cytoplasm and membrane in LPS-treated THP-1 cells. We found that Vangl2 interacted with p65 mainly in the cytoplasm, although most of Vangl2 located on the membrane (*Figure 3—figure supplement 1L*). Together, these data suggest that Vangl2 may interact with p65 in the cytoplasm to inhibit NF-κB signaling.

Vangl2 comprises an N-terminal cytoplasmic tail (NT), a transmembrane (TM) domain, a Prickle-binding domain (PkBD), and a C-terminal cytoplasmic tail (CT) (*Nagaoka et al., 2019*). To map the essential domains of Vangl2 that mediate its association with p65, we generated several deletion constructs of Vangl2. We found Vangl2 FL (full-length), T2 (ΔNT + TM), T3 (ΔCT), and T4 (ΔNT) interacted with the full-length p65, while Vangl2 T1 (ΔPkBD + CT) abrogated their association (*Figure 3H* and *Figure 3—figure supplement 1M*), indicating that the PkBD domain is important for the Vangl2–p65 interaction. Additionally, we constructed HA-tagged Vangl2 PkBD plasmid and co-IP assay revealed that Vangl2 FL and PkBD interacted with p65 (*Figure 3I*). Moreover, we observed that

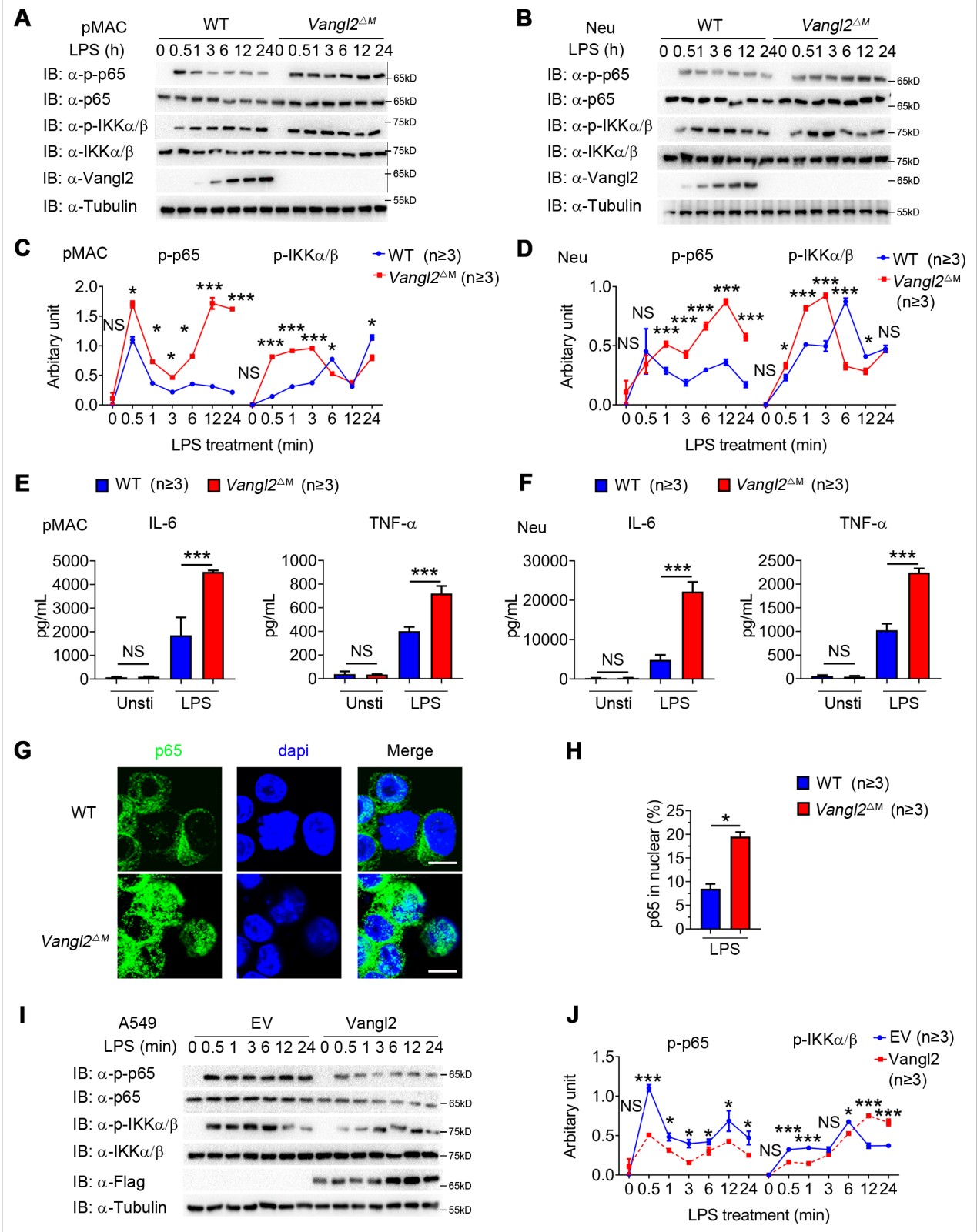

**Figure 2.** Van Gogh-like 2 (Vangl2) negatively regulates lipopolysaccharide (LPS)-induced NF-$\kappa$B activation and proinflammatory cytokines. Wild-type (WT) and Vangl2-deficient ($n \geq 3$) pMAC (**A, C**) or neutrophils (**B, D**) were stimulated with LPS (100 ng/ml) for the indicated times. Immunoblot analysis of total and phosphorylated p65, IKK$\alpha/\beta$ (**A, B**), and analysis of gray intensity was shown (**C, D**) ($n \geq 3$). (**E, F**) WT and Vangl2-deficient ($n \geq 3$) pMAC or neutrophils were stimulated with LPS (100 ng/ml) for 6 hr. mRNA levels of *Il6* and *Tnfa* were measure by qPCR (**E**). IL-6 and tumor necrosis factor-$\alpha$

*Figure 2 continued on next page*

*Figure 2 continued*

(TNF-α) secretion by WT and Vangl2-deficient bone marrow-derived macrophages (BMDMs) or neutrophils treated with or without LPS for 6 hr was measured by enzyme-linked immunosorbent assay (ELISA) (**F**) ($n \geq 3$). (**G, H**) The WT and Vangl2-deficient ($n \geq 3$) neutrophils were treated with LPS (1000 ng/ml) for 4 hr, and the nuclear translocation of p65 was detected by immunofluorescence (**G**) (p65, green; 4'-6-diamidino-2-phenylindole (DAPI), blue; scale bar, 50 µm). Percentages of p65 nuclear translocated cells in WT and Vangl2-deficient neutrophils were determined by counting 100–150 cells in non-overlapping fields (**H**) ($n \geq 3$). (**I, J**) A549 cells were transfected with Flag-tagged Vangl2 plasmid or empty vector, then stimulated with LPS (100 ng/ml) for the indicated times. Immunoblot analysis of total and phosphorylated p65, IKKα/β (**I**) (representative image), and analysis of gray intensity was shown (**J**) ($n \geq 3$). pMAC, peritoneal macrophage; Neu, neutrophil; EV, empty vector. Data are representative of three independent experiments and are plotted as the mean ± standard deviation (SD). Multiple t tests for C, D, E, F and J. Two-tailed Student's t test for H. *$p < 0.05$, ***$p < 0.001$ vs. corresponding control.

The online version of this article includes the following source data and figure supplement(s) for figure 2:

**Source data 1.** Uncropped and labelled gels for *Figure 2*.

**Source data 2.** Raw data for *Figure 2*.

**Source data 3.** Raw unedited gels for *Figure 2*.

**Figure supplement 1.** Van Gogh-like 2 (Vangl2) defection promotes lipopolysaccharide (LPS)-induced NF-κB activation and production of inflammatory cytokines.

**Figure supplement 1—source data 1.** Raw data for (*Figure 2—figure supplement 1*).

**Figure supplement 1—source data 2.** Raw data for (*Figure 2—figure supplement 1*).

**Figure supplement 1—source data 3.** Raw unedited gels for (*Figure 2—figure supplement 1*).

deletion of the PkBD domain (T1), but not other domains, of Vangl2 abolished the Vangl2-mediated inhibition of NF-κB-luc activity, and transfection of Vangl2 PkBD domain achieved similar inhibition on p65 induced NF-κB activation as Vangl2 FL did (*Figure 3J*). Collectively, these data suggest that Vangl2 suppresses NF-κB signaling by targeting p65 through Vangl2 PkBD domain interaction.

## Vangl2 promotes the autophagic degradation of p65

Next we sought to study the physiological role of Vangl2-p65 interaction in the regulation of NF-κB signaling by transfecting 293T cells with Flag-tagged p65, together with increasing doses of Vangl2, and found that Vangl2 dramatically decreased the protein level of p65 in a dose-dependent manner (*Figure 4A*). To exclude the possibility that the downregulation of p65 protein by Vangl2 was caused by the inhibition of p65 transcription, qPCR results suggested that the abundance of *p65* mRNA did not change in cells with increased expression of Vangl2 (*Figure 4B*). Moreover, Vangl2 facilitated the degradation of the p65 phosphorylation mutants (*Figure 4—figure supplement 1A*), and Vangl2 phosphorylation mutants were observed to degrade p65 (*Figure 4—figure supplement 1B*), indicating that Vangl2-mediated p65 degradation is independent of its phosphorylation status. Since activated p65 translocates to the nucleus, we next assessed whether Vangl2 regulates the degradation of p65 in the cytoplasm or nucleus, and found that the Vangl2 interacted with p65 mainly in the cytoplasm (*Figure 4—figure supplement 1C*) and mediated the degradation of p65 in the cytoplasmic fraction (*Figure 4C*), which is consistent with the result that Vangl2 and p65 co-localized in the cytoplasm (*Figure 3G*). To further assess whether Vangl2 regulates the degradation of endogenous p65, we found that Vangl2-deficient BMDMs stabilized the expression of endogenous p65 after LPS treatment (*Figure 4D*). After 12 hr of LPS stimulation, the cycloheximide (CHX)-chase assay result showed that the degradation rate of p65 in *Vangl2*^ΔM BMDMs was slower than that of WT cells (*Figure 4—figure supplement 1D*). Together, these data suggest that Vangl2 promotes p65 protein degradation.

To investigate whether Vangl2 degrades p65 through an autolysosome or proteasome pathway, 293T cells were transfected with p65, together with or without the Vangl2 plasmids, and treated with different pharmacological inhibitors. We found the degradation of p65 induced by Vangl2 was blocked by autolysosome inhibitor chloroquine (CQ) and bafilomycin A1 (Baf A1), or autophagy inhibitor 3-methyladenine (3-MA) (*Figure 4E*), but not by the proteasome inhibitor MG132 or caspase-1 inhibitor Z-VAD and VX-765 (*Figure 4E* and *Figure 4—figure supplement 1E*). Furthermore, Vangl2 significantly increased the degradation of p65 during rapamycin-triggered autophagy (*Figure 4F*), which suggested that Vangl2 promoted autophagic degradation of p65.

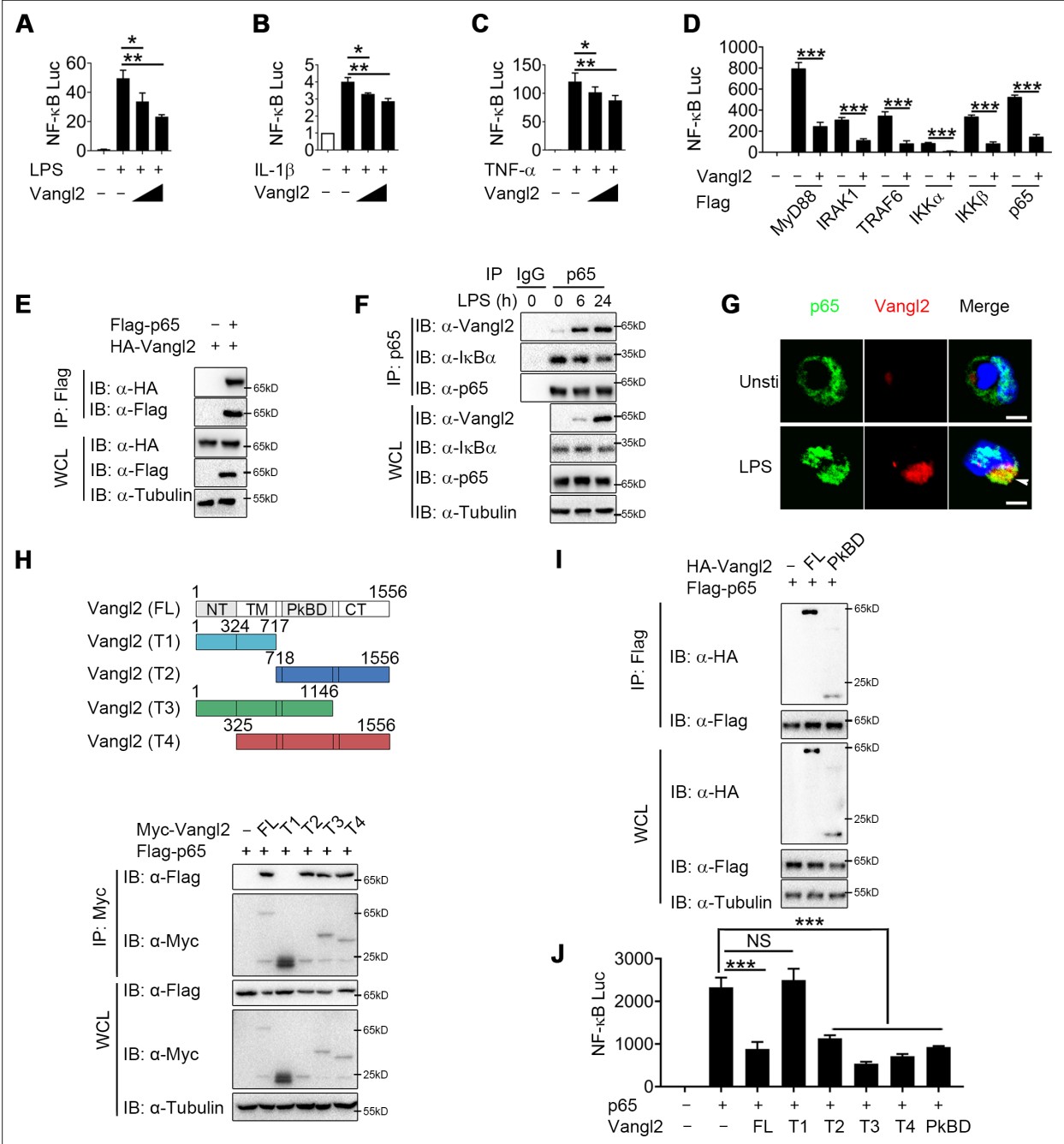

**Figure 3.** Van Gogh-like 2 (Vangl2) inhibits NF-κB signaling by interacting with p65. Cho (**A**) or HEK293T cells (**B, C**) were co-transfected with a NF-κB and TK-Renilla reporter along with increasing amounts of Vangl2 for 18 hr, then treated the cells with or without lipopolysaccharide (LPS) (**A**, 250 ng/ml), IL-1β (**B**, 40 ng/ml), or tumor necrosis factor-α (TNF-α) (**C**, 20 ng/ml) for 6 hr. NF-κB promoter-driven luciferase activity was measured and normalized to the Renilla luciferase activity (n ≥ 3). (**D**) Luciferase activity in HEK293T transfected with plasmids encoding an NF-κB luciferase reporter and TK-Renilla reporter, together with a vector encoding MyD88, IRAK1, TRAF6, IKKα, IKKβ, or p65, along with or without Vangl2 plasmid, was measured at 24 hr after transfection and normalized to the Renilla luciferase activity (n ≥ 3). (**E**) HEK293T cells were transfected with plasmids encoding HA-tagged Vangl2 and Flag-tagged p65, followed by immunoprecipitation with anti-Flag beads and immunoblot analysis with anti-HA. Throughout was the immunoblot analysis of whole-cell lysates without immunoprecipitation. (**F**) Bone marrow-derived macrophages (BMDMs) were stimulated with LPS (100 ng/ml) for the indicated times. The cell lysates were subjected to immunoprecipitation with an anti-p65 antibody or control IgG, followed by immunoblotting with indicated antibodies. (**G**) The wild-type (WT) and Vangl2-deficient peritoneal macrophages were treated with LPS (1000 ng/ml) for 4 hr, and co-localization of p65 and Vangl2 was detected by immunofluorescence (p65, green; Vangl2, red; DAPI, blue; scale bar, 50 μm). (**H**) A structural diagram of Vangl2 as well as schematic representation of Myc-tagged truncation mutants of Vangl2 (top). HEK293T cells were transfected with Flag-tagged p65 and empty vector, Myc-tagged Vangl2 (FL) or Vangl2 truncation mutants. The cell lysates were subjected to immunoprecipitation with anti-Flag beads

*Figure 3 continued on next page*

*Figure 3 continued*

and immunoblotted with the indicated antibodies (bottom). (**I**) HEK293T cells were transfected with Flag-tagged p65 and HA-tagged Vangl2 FL or PkBD truncation. The cell lysates were subjected to immuprecipitation with anti-Flag beads and immunoblotted with the indicated antibodies. (**J**) Luciferase activity in HEK293T cells transfected with an NF-$\kappa$B luciferase reporter, together with a vector encoding p65, along with the empty vector or with vectors encoding Vangl2 or its truncation mutants ($n \geq 3$). The results are presented relative to Renilla luciferase activity. IP, immunoprecipitation; WCL, whole-cell lysate. Data are representative of three independent experiments and are plotted as the mean ± standard deviation (SD). Multiple t tests for A-D, and J. *p < 0.05, **p < 0.01, ***p < 0.001 vs. corresponding control. NS, not significant.

The online version of this article includes the following source data and figure supplement(s) for figure 3:

**Source data 1.** Uncropped and labelled gels for *Figure 3*.

**Source data 2.** Raw data for *Figure 3*.

**Source data 3.** Raw unedited gels for *Figure 3*.

**Figure supplement 1.** Van Gogh-like 2 (Vangl2) interacts with p65 to inhibit NF-$\kappa$B activation.

**Figure supplement 1—source data 1.** Uncropped and labelled gels for (*Figure 3—figure supplement 1*).

**Figure supplement 1—source data 2.** Raw data for (*Figure 3—figure supplement 1*).

**Figure supplement 1—source data 3.** Raw unedited gels for (*Figure 3—figure supplement 1*).

To further demonstrate Vangl2-mediated p65 degradation through autophagy, we transfected WT, ATG5, and Beclin1 knockout (KO) 293T cells with Vangl2, and found that the degradation of p65 triggered by Vangl2 was almost abrogated in ATG5 and Beclin1 KO cells (*Figure 4G*). The p65 turnover rates were markedly reduced in ATG5 and Beclin1 KO cells post CHX treatment (*Figure 4—figure supplement 1F, G*), suggesting that the impaired autophagy prevented p65 degradation. Consistently, NF-κB activation induced by p65 was rescued in ATG5 and Beclin1 KO cells expressing Vangl2, compared to WT 293T cells (*Figure 4H, I*). In addition, the co-localization of p65 and LC3B was enhanced following LPS stimulation (*Figure 4—figure supplement 1H*). Together, our data suggest that Vangl2 specifically promotes p65 degradation through autophagy.

## Vangl2 promotes the recognition of p65 by cargo receptor NDP52

Accumulating evidence showed that cargo receptors play crucial roles in selective autophagic degradation by delivering substrates (*He et al., 2019*; *Kirkin and Rogov, 2019*; *Wu et al., 2021*). Since there is no research suggests that Vangl2 is a cargo protein, we hypothesized that Vangl2 might bridge p65 to the cargo receptors for autophagic degradation. To identify the potential cargo receptor responsible for Vangl2-mediated autophagic degradation of p65, we co-transfected 293T cells with Vangl2 and various cargo receptors, followed by co-IP assay. Result suggested that Vangl2 strongly interacted with the cargo receptors p62 and NDP52, and slightly associated with NBR1 and Nix (*Figure 5A*). However, p65 only interacted with the cargo receptors p62 and NDP52 (*Figure 5B*), which is consistent with a recent finding that p62 protein is a Vangl2-binding partner (*Puvirajesinghe et al., 2016*). We next attempted to clarify whether p62 or NDP52 is involved in Vangl2-mediated autophagic degradation of p65. Interestingly, we found that Vangl2 promoted the association between p65 and NDP52, but did not affect p65–p62 complex (*Figure 5C*). Consistently, we found that Vangl2-mediated degradation of p65 was rescued in NDP52 KO cells, but not in p62 KO cells (*Figure 5D*). Endogenous co-IP immunoblot analyses also revealed that deficiency of Vangl2 remarkably attenuated the association of endogenous p65 and NDP52 (*Figure 5E*). Likewise, NDP52, but not p62, enhanced the association between p65 and Vangl2 (*Figure 5F* and *Figure 5—figure supplement 1A*). And Vangl2 failed to inhibit the activation of NF-κB signaling in NDP52 KO cells (*Figure 5G*), but not in p62 KO cells (*Figure 5—figure supplement 1B*). Furthermore, compared with WT cells, CHX-chase assay results showed that the degradation rates of p65 were reduced in NDP52 KO cells (*Figure 5H*), but not in p62 KO cells (*Figure 5—figure supplement 1C*). Taken together, these data suggest that Vangl2 mediates the NDP52-directed selective autophagic degradation of p65.

## Vangl2 increases the K63-linked poly-ubiquitination of p65

It has been well documented that ubiquitin chains attached to the substrates and served as a signal for the recognition by cargo receptors (*Otten et al., 2021*; *Shaid et al., 2013*; *Yin et al., 2020*). The ubiquitin-associated (UBA) domain of NDP52 mostly recognized ubiquitinated substrates for degradation through autophagy (*Johansen and Lamark, 2011*; *Yamano and Youle, 2020*). We hypothesized

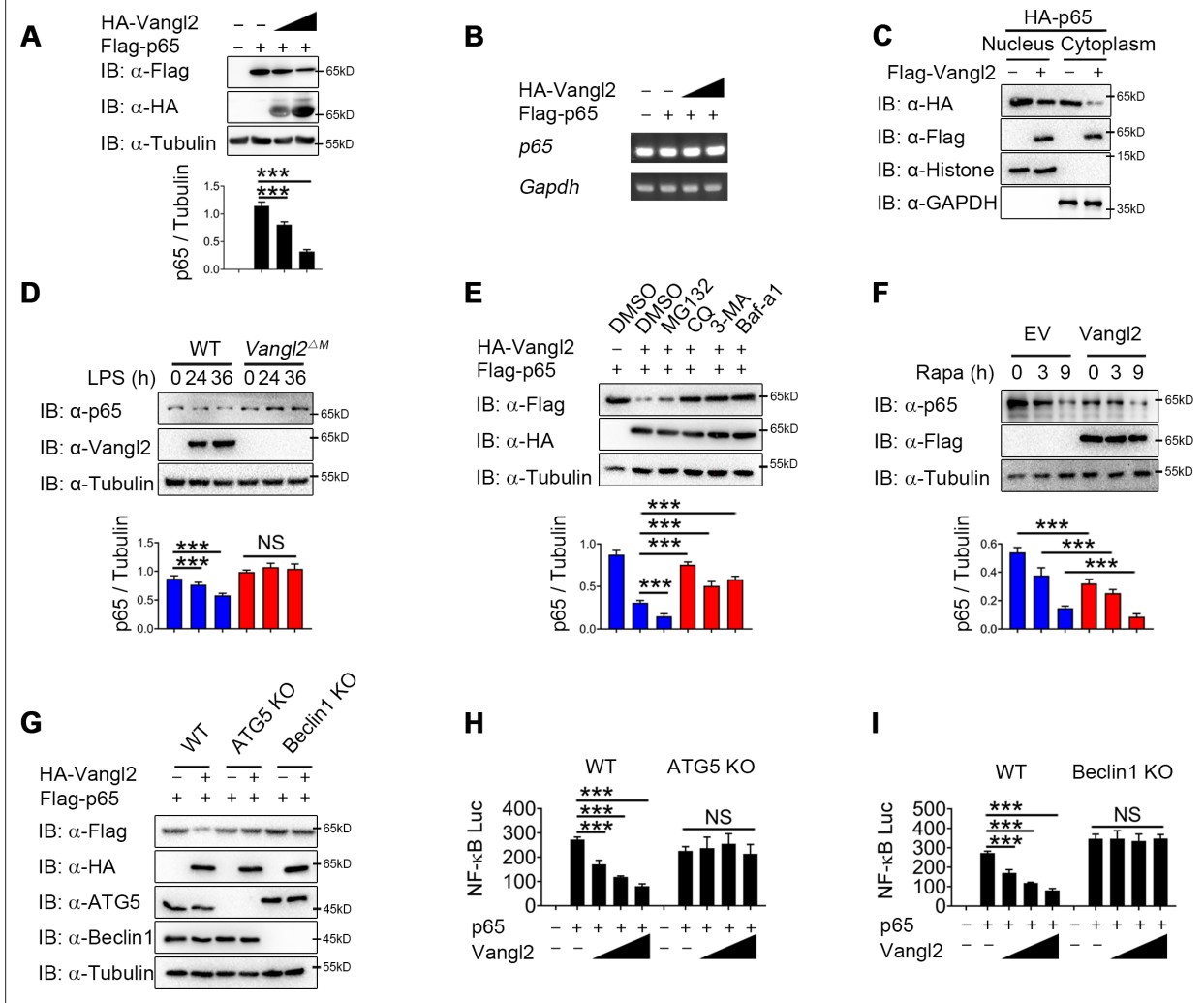

**Figure 4.** Van Gogh-like 2 (Vangl2) promotes the autophagic degradation of p65. (**A**) Immunoblot analysis of HEK293T cells transfected with Flag-p65 and increasing amounts of the vector encoding HA-Vangl2 (0, 250, 500, and 1000 ng) ($n \geq 3$). (**B**) Total RNA from HEK293T cells as in (**A**) was isolated and measured by semi-quantitative PCR. (**C**) HEK293T cells transfected with Flag-p65 and increasing amounts of the vector encoding HA-Vangl2, and the expressions of p65 in nuclear or cytoplasm were detected by immunoblot. (**D**) Wild-type (WT) and Vangl2-deficient bone marrow-derived macrophages (BMDMs) were treated with lipopolysaccharide (LPS) for the indicated times, and the expressions of p65 and Vangl2 were detected by immunoblot ($n \geq$ 3). (**E**) HEK293T cells were transfected with Flag-p65 and HA-Vangl2 plasmids, and treated with DMSO, MG132 (10 μM), CQ (50 μM), 3-MA (10 mM), or Baf-A1 (0.2 μM) for 6 hr. The cell lysates were analyzed by immunoblot ($n \geq 3$). (**F**) HEK293T cells were transfected with empty vector (EV) or Flag-Vangl2 plasmid, and treated with rapamycin for the indicated times. The cell lysates were analyzed by immunoblot with indicated antibodies ($n \geq 3$). (**G**) WT, ATG5 knockout (KO), and Beclin1 KO HEK293T cells were transfected with Flag-p65, together with or without HA-Vangl2 plasmids, and then the cell lysates were analyzed by immunoblot with indicated antibodies. Luciferase activity in WT, ATG5 KO (**H**) and Beclin1 KO (**I**) HEK293T cells transfected with plasmids encoding an NF-κB luciferase reporter and TK-Renilla reporter, together with p65 plasmid along with increasing amounts of Vangl2, was measured at 24 hr after transfection and normalized to the Renilla luciferase activity ($n \geq 3$). CHX, cycloheximide; 3-MA, 3-methyladenine; CQ, chloroquine; Baf A1, bafilomycin A1. Data are representative of three independent experiments and are plotted as the mean ± standard deviation (SD). Multiple t tests for A, D, E, F, H and I. ***$p < 0.001$ vs. corresponding control. NS, not significant.

The online version of this article includes the following source data and figure supplement(s) for figure 4:

**Source data 1.** Uncropped and labelled gels for *Figure 4*.

**Source data 2.** Raw data for *Figure 4*.

**Source data 3.** Raw unedited gels for *Figure 4*.

**Figure supplement 1.** Van Gogh-like 2 (Vangl2) promotes p65 degradation by autophagic pathway.

**Figure supplement 1—source data 1.** Uncropped and labelled gels for (*Figure 4—figure supplement 1*).

**Figure supplement 1—source data 2.** Raw data for (*Figure 4—figure supplement 1*).

**Figure supplement 1—source data 3.** Raw unedited gels for (*Figure 4—figure supplement 1*).

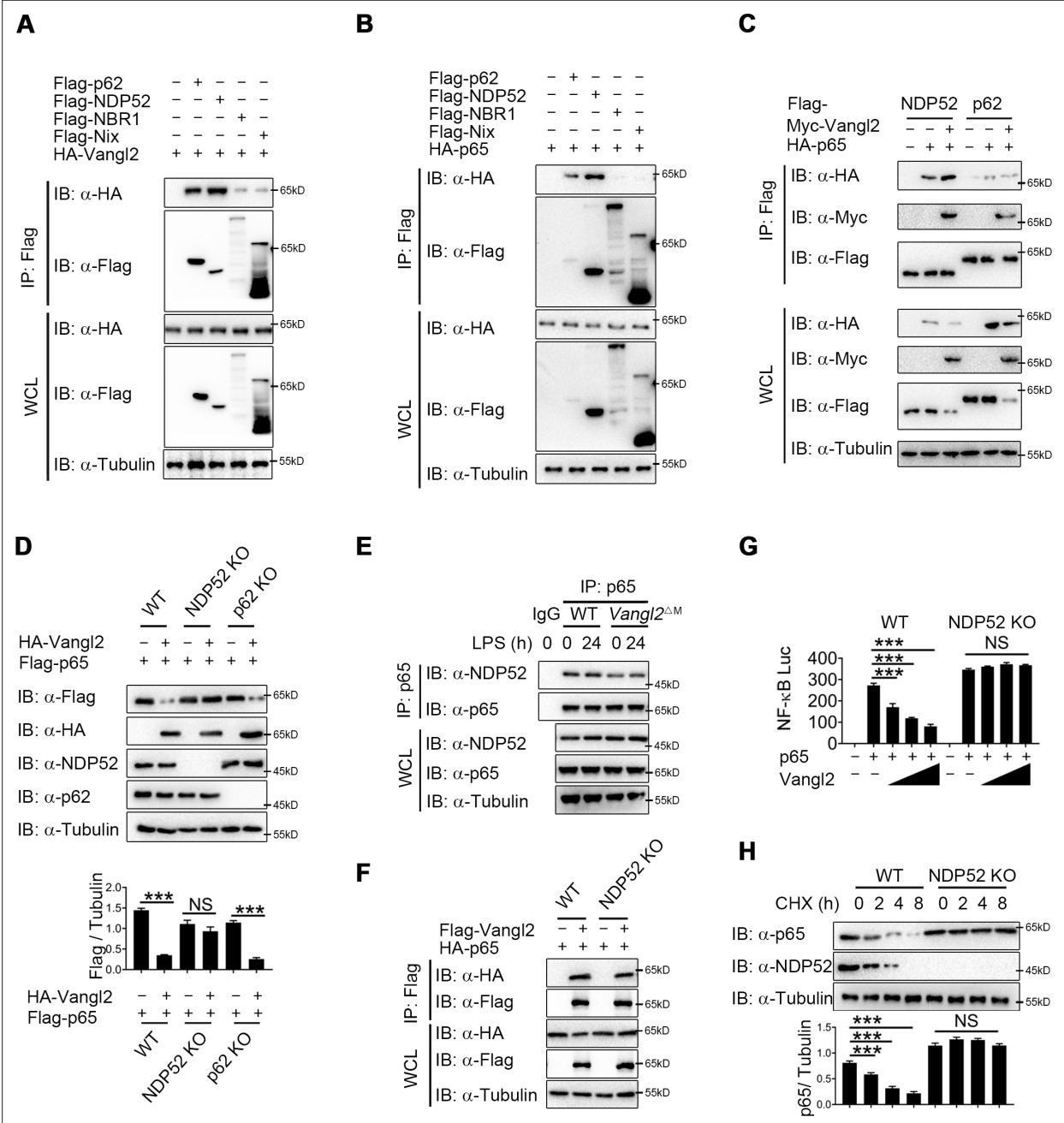

**Figure 5.** Van Gogh-like 2 (Vangl2) enhances the recognition of p65 by cargo receptor NDP52. (**A**) HEK293T cells transfected with a vector expressing HA-Vangl2 along with the empty vector or vector encoding Flag-p62/NDP52/NBR1/Nix. The cell lysates were subjected to immunoprecipitation with anti-Flag beads and immunoblotted with the indicated antibodies. (**B**) HEK293T cells transfected with a vector expressing HA-p65 along with the empty vector or vector encoding Flag-p62/NDP52/NBR1/Nix. The cell lysates were subjected to immunoprecipitation with anti-Flag beads and immunoblotted with the indicated antibodies. (**C**) HEK293T cells were transfected with HA-p65 together with Flag-NDP52 or Flag-p62, as well as with empty vector or Myc-Vangl2. The cell lysates were subjected to immunoprecipitation with anti-Flag beads and immunoblotted with the indicated antibodies. (**D**) Wild-type (WT), NDP52 knockout (KO), and p62 KO HEK293T cells were transfected with a vector expressing HA-p65 along with the empty vector or vector encoding Flag-Vangl2. The cell lysates were immunoblotted with the indicated antibodies ($n \geq 3$). (**E**) WT and Vangl2-deficient bone marrow-derived macrophages (BMDMs) were stimulated with lipopolysaccharide (LPS) (100 ng/ml) for the indicated times. The cell lysates were subjected to immunoprecipitation with an anti-p65 antibody or control IgG and immunoblotted with the indicated antibodies. (**F**) WT and NDP52 KO HEK293T cells were transfected with a vector expressing HA-p65 along with the empty vector or vector encoding Flag-Vangl2. The cell lysates were subjected to immunoprecipitation with anti-Flag beads and immunoblotted with the indicated antibodies. (**G**) Luciferase activity in WT and NDP52 KO HEK293T cells transfected with plasmids encoding NF-$\kappa$B luciferase reporter and TK-Renilla reporter, together with p65 plasmid along with increasing amounts of Vangl2 plasmid, was measured at 24 hr after transfection ($n \geq 3$). (**H**) WT and NDP52 KO HEK293T were treated with cycloheximide (CHX) for the

*Figure 5 continued on next page*

*Figure 5 continued*

indicated times. The cell lysates were immunoblotted with the indicated antibodies (*n* ≥ 3). Data are representative of three independent experiments and are plotted as the mean ± standard deviation (SD). Multiple t tests for D, G and H. ***p < 0.001 vs. corresponding control. NS, not significant.

The online version of this article includes the following source data and figure supplement(s) for figure 5:

**Source data 1.** Uncropped and labelled gels for *Figure 5*.

**Source data 2.** Raw data for *Figure 5*.

**Source data 3.** Raw unedited gels for *Figure 5*.

**Figure supplement 1.** Van Gogh-like 2 (Vangl2) promoted autophagic degradation of p65 is not mediated by cargo receptor p62.

**Figure supplement 1—source data 1.** Uncropped and labelled gels for (*Figure 5—figure supplement 1*).

**Figure supplement 1—source data 2.** Raw data for (*Figure 5—figure supplement 1*).

**Figure supplement 1—source data 3.** Raw unedited gels for (*Figure 5—figure supplement 1*).

that Vangl2 may affect the ubiquitin chains on p65 for subsequent NDP52-dependent degradation. To demonstrate this, we performed endogenous co-IP assay and found that the poly-ubiquitination of p65 was impaired in BMDMs from *Vangl2*$^{\Delta M}$ mice after LPS stimulation, compared to WT mice (*Figure 6A*). Moreover, Vangl2 specifically increased the total and K63-linked (K63-only ubiquitin mutant) poly-ubiquitination of p65 (*Figure 6B*). In contrast, Vangl2 did not affect other ubiquitin linkages (K11, K27, K33, or K48-only ubiquitin mutant) of p65 in an overexpression system (*Figure 6C*). To further substantiate the hypothesis that Vangl2 is involved in K63-linked poly-ubiquitinated p65, we conducted a knockdown experiment using small interfering RNA (siRNA) to reduce Vangl2 expression. Our results demonstrated that the absence of Vangl2 inhibits K63-linked poly-ubiquitination of p65 (*Figure 6D*). Together, these results suggest that Vangl2 promotes the degradation of p65 by promoting the K63-linked ubiquitination of p65.

## Vangl2 recruits PDLIM2 to ubiquitinate p65

Although Vangl2 promotes K63-linked ubiquitination and degradation of p65, Vangl2 is not an E3 ubiquitin ligase. We hypothesized that Vangl2 might function as a scaffold protein to link p65 and its E3 ubiquitin ligase for ubiquitination or to block the interaction of deubiquitinase (DUB) with p65. To identify the potential E3 ubiquitinase or DUB responsible for Vangl2-mediated ubiquitination of p65, we further analyzed our RNA-seq data to identify E3 ubiquitin ligase or DUB involved in ubiquitination of p65 by comparing LPS-stimulated BMDMs from *Vangl2*$^{\Delta M}$ and WT mice. The result of differentially expression identified 88 downregulated genes related to E3 ubiquitin ligase and 56 upregulated genes related to DUB in response to Vangl2 deficiency in BMDMs after LPS stimulation (*Figure 7—figure supplement 1A*), which included PDZ-LIM domain-containing protein 2 (PDLIM2), Trim21, and DUB ubiquitin-specific peptidase 7 (USP7). Recent research showed that ubiquitin E3 ligases PDLIM2 and Trim21 ubiquitinated p65 via K63 linkage and enhanced the interaction of p65 with IKK (*Healy and O'Connor, 2009*; *Jodo et al., 2020*; *Yang et al., 2021*), while DUB USP7 promoted NF-κB-mediated transcription (*Mitxitorena et al., 2020*). Here, we found that the mRNA level of PDLIM2 and USP7 decreased in Vangl2-deficient BMDMs, compared with WT BMDMs (*Figure 7—figure supplement 1B–D*). To investigate which E3 ubiquitin ligase or DUB is recruited by Vangl2, luciferase assay in Vangl2-expressing 293T cells transfected with scramble, *Pdlim2*, *Usp7*, or *Trim21* siRNA (*Figure 7—figure supplement 1E–G*) suggested Vangl2-mediated inhibition of NF-κB activation and degradation of p65 were blocked by knocking down PDLIM2, but not USP7 or Trim21 (*Figure 7A*). Meanwhile, *Pdlim2* knockdown in BMDMs also resulted in higher expression of *Il6* and *Il1b* in response to LPS stimulation (*Figure 7B, C*), indicating that PDLIM2 plays a key role in promoting Vangl2-mediated p65 degradation.

We next investigated whether Vangl2 promoted the association between p65 and PDLIM2 by co-IP experiments. Our results indicated that Vangl2 interacted with PDLIM2 (*Figure 7D*) and promoted the association between p65 and PDLIM2 (*Figure 7E*). Notably, PDLIM2 accelerated the degradation of p65 in the presence of Vangl2 (*Figure 7F*). Conversely, PDLIM2 deficiency markedly impaired Vangl2-mediated K63-linked ubiquitination of p65 (*Figure 7G*) and knockdown of Vangl2 inhibited K63-linked poly-ubiquitination of p65 mediated by PDLIM2 (*Figure 7H*). Taken together, these data suggest that

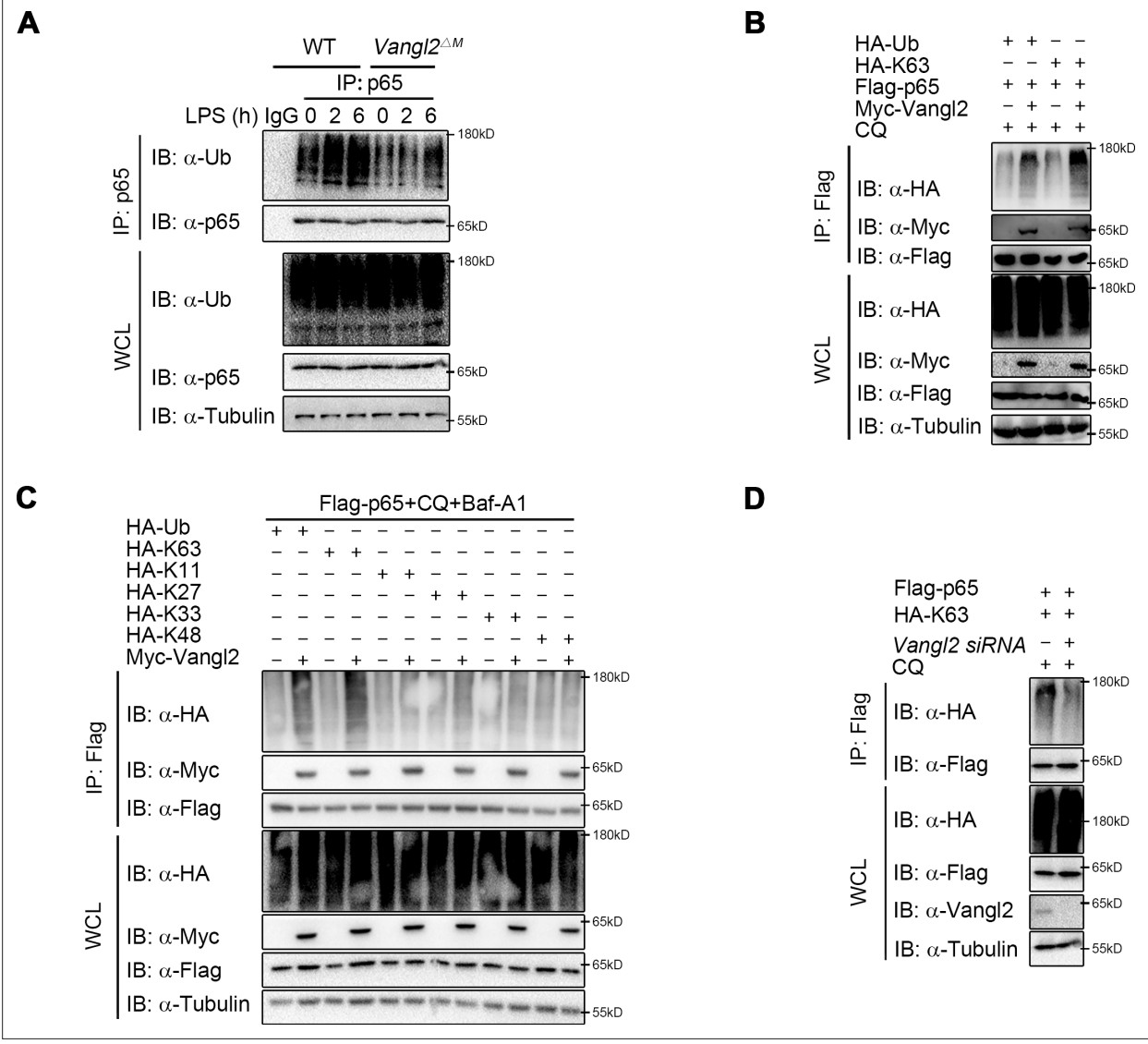

**Figure 6.** Van Gogh-like 2 (Vangl2) increases the K63-linked ubiquitination of p65. (**A**) Wild-type (WT) and Vangl2-deficient bone marrow-derived macrophages (BMDMs) were stimulated with lipopolysaccharide (LPS) (100 ng/ml) for the indicated times. The cell lysates were subjected to immunoprecipitation with an anti-p65 antibody or control IgG and immunoblotted with the indicated antibodies. (**B**) HEK293T cells were transfected with Flag-p65, Myc-Vangl2, HA-Ub, or HA-K63 plasmids with the indicated combinations for 24 hr and then treated with chloroquine (CQ) for 8 hr. The cell lysates were subjected to immunoprecipitation with anti-Flag beads and immunoblotted with the indicated antibodies. (**C**) HEK293T cells were transfected with Flag-p65, Myc-Vangl2, and HA-Ub/K63/K11/K27/K33/K48 plasmids with the indicated combinations for 24 hr and then treated with CQ and Baf-A1 for 8 hr. The cell lysates were subjected to immunoprecipitation with anti-Flag beads and immunoblotted with the indicated antibodies. (**D**) HEK293T cells were transfected with a vector expressing Flag-p65 and HA-K63 along with *Scramble* or *Vangl2* small interfering RNA (siRNA). The cell lysates were subjected to immunoprecipitation with anti-Flag beads and immunoblotted with the indicated antibodies. Data are representative of three independent experiments.

The online version of this article includes the following source data for figure 6:

**Source data 1.** Uncropped and labelled gels for *Figure 6*.

**Source data 2.** Raw unedited gels for *Figure 6*.

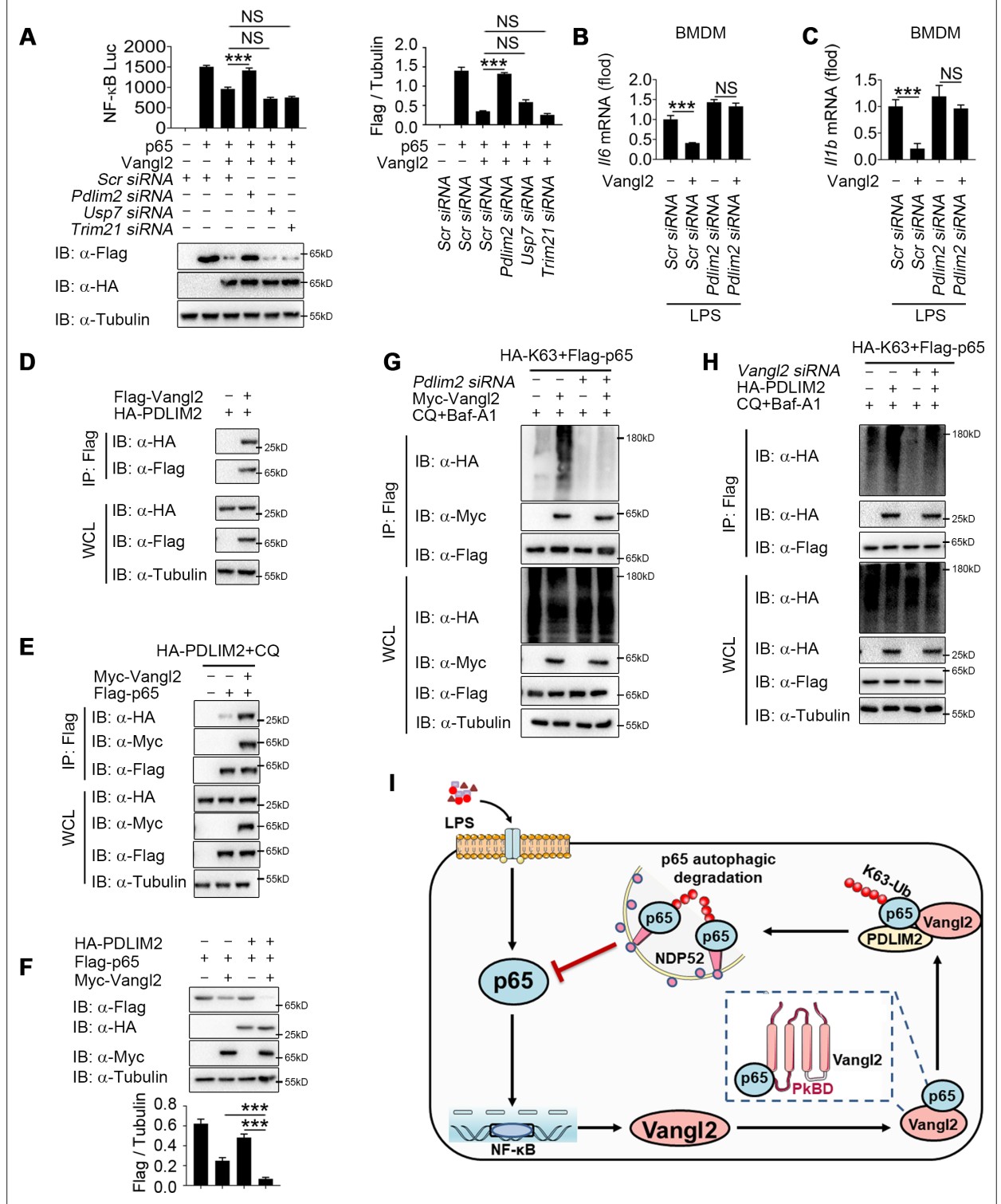

**Figure 7.** Van Gogh-like 2 (Vangl2) recruits PDLIM2 to ubiquitinate p65. (**A**) HEK293T cells were transfected with the indicated small interfering RNA (siRNA), NF-κB reporter plasmids together with HA-Vangl2, Flag-p65, or the control vector as indicated for 24 hr, and then subjected to luciferase assay and immunoblotting analysis (*n* ≥ 3). Bone marrow-derived macrophages (BMDMs) were transfected with *Pdlim2* or *Scramble* siRNA along with the empty vector or vector encoding Flag-Vangl2, stimulated with lipopolysaccharide (LPS) (100 ng/ml) for 6 hr, then analyzed by qPCR for *Il6* (**B**) and *Il1b* (**C**) expression (*n* ≥ 3). (**D**) HEK293T cells transfected with HA-PDLIM2 along with the empty vector or vector encoding Flag-Vangl2. The cell lysates were subjected to immunoprecipitation with anti-Flag beads and immunoblotted with the indicated antibodies. (**E**) HEK293T cells were transfected with Flag-p65, HA-PDLIM2 and Myc-Vangl2 plasmids with the indicated combinations for 24 hr and then treated with chloroquine (CQ) and Baf-A1 for 8 hr.

*Figure 7 continued on next page*

*Figure 7 continued*

The cell lysates were subjected to immunoprecipitation with anti-Flag beads and immunoblotted with the indicated antibodies. (**F**) HEK293T cells were transfected with Flag-p65, HA-PDLIM2, and Myc-Vangl2 plasmids with the indicated combinations for 24 hr. The cell lysates were immunoblotted with the indicated antibodies ($n \geq 3$). (**G**) HEK293T cells were transfected with Flag-p65, HA-K63, and Myc-Vangl2 plasmids, the expression of E3 ubiquitin ligase was interfered with *Pdlim2* siRNA and then treated with CQ and Baf-A1 for 8 hr. The cell lysates were subjected to immunoprecipitation with anti-Flag beads and immunoblotted with the indicated antibodies. (**H**) HEK293T cells were transfected with *Pdlim2* or *Scramble* siRNA, along with or without HA-PDLIM2, then treated with CQ and Baf-A1 for 8 hr. The cell lysates were subjected to immunoprecipitation with anti-Flag beads and immunoblotted with the indicated antibodies. (**I**) A schematic model to illustrate how Vangl2-PDLIM2-NDP52-p65 axis negatively regulates NF-$\kappa$ B activation. During LPS stimulation, Vangl2 expression is upregulated, thus constituting a negative feedback loop to regulate NF-$\kappa$ B activation. In detail, Vangl2 functions as an adaptor protein to recruit an E3 ubiquitin ligase PDLIM2 to increase K63-linked ubiquitination of p65 and promotes NDP52-mediated p65 degradation through selective autophagy, resulting in ameliorating sepsis and suppressing production of proinflammatory cytokines. Data are representative of three independent experiments and are plotted as the mean ± standard deviation (SD). Multiple t tests for A-C, and F. ***$p < 0.001$ vs. corresponding control. NS, not significant.

The online version of this article includes the following source data, source code, and figure supplement(s) for figure 7:

**Source data 1.** Uncropped and labelled gels for *Figure 7*.

**Source data 2.** Raw unedited gels for *Figure 7*.

**Source data 3.** Raw data for *Figure 7*.

**Figure supplement 1.** Expression of candidate E3 ubiquitin ligases in wild-type (WT) and Van Gogh-like 2 (Vangl2)-deficient bone marrow-derived macrophages (BMDMs) after lipopolysaccharide (LPS) treatment.

**Figure supplement 1—source code 1.** Raw data for *Figure 7—figure supplement 1*.

Vangl2 functions as a potential adaptor to recruit E3 ubiquitin ligase PDLIM2 to p65 and promotes the K63-linked ubiquitination of p65 for its subsequent autophagic degradation.

## Discussion

As a core PCP component, Vangl2 is widely known for its function in organ development, such as brain, tooth, tongue, and kidney (*Bailly et al., 2018*; *Hatakeyama et al., 2014*), trafficking from the ER to the cell surface, and subsequently shuttles between the endocytic vesicles and cell surface (*Feng et al., 2021*). The function of Vangl2 mostly depends on its cellular localization (*Hatakeyama et al., 2014*) and is reported mainly through PCP/WNT signaling pathway (*Jessen and Jessen, 2019*). Activated Vangl2 exhibits extremely long cytoplasmic and intercellular branches and delivers Wnt to multiple cells to enhance Wnt/β-catenin signaling (*Brunt et al., 2021*). During myocardial hypertrophy, Vangl2 aggravates myocardial hypertrophy by regulating Wnt/c-Jun N-terminal kinase (JNK) signaling (*Brunt et al., 2021*; *Jessen and Jessen, 2017*; *Jessen and Jessen, 2019*) and the expansion of cardiomyocyte surface area (*Brunt et al., 2021*; *Jessen and Jessen, 2017*; *Jessen and Jessen, 2019*). However, our previous study showed that the lysosome-suppressing function of Vangl2 in osteoblast differentiation is not dependent on conventional PCP pathway (*Gong et al., 2021*), and we found Vangl2 did not affect the JNK pathway after LPS treatment (data not shown), suggesting that Vangl2 has additional functions in pathways besides PCP. In this study, we provided direct evidence that the expression of Vangl2 was increased during sepsis and upregulated significantly in immune organs (lymph nodes and spleen) upon LPS stimulation, which is consistent with our previous finding that Vangl2 regulated the downstream signaling of TLR or IL-1R. In addition, the present study showed that Vangl2 prevented the progression of sepsis and the accumulation of inflammatory cytokines through suppressing NF-κB pathway: Vangl2 inhibited LPS-induced NF-κB activation by delivering p65 to autophagosome for degradation. To the best of our knowledge, this is the first study proves that Vangl2 regulates NF-κB signaling and inflammatory responses, and could serve as a potential target for therapeutic purposes in diseases associated with NF-κB signaling.

As is well-known, Vangl2 phosphorylation is essential for its functions in the mammalian PCP pathway in multiple tissues. However, our findings indicate that Vangl2 phosphorylation mutants also promote p65 degradation (*Figure 4—figure supplement 1B*) and the degradation of p65 induced by Vangl2 was blocked by autolysosome inhibitor (CQ), but not by the JNK inhibitor (SP600125) or Wnt/β-catenin inhibitor (FH535) (data no shown), suggesting that Vangl2 may regulate the NF-κB pathway independently of its conventional PCP pathway. Nevertheless, additional pathway inhibitions, such as those of the HH/GLI and Fat-Dachsous pathways, should also be employed to further

elucidate the function of Vangl2 in p65 degradation. The PCP pathways frequently intersect with developmentally important pathways. Previous research has demonstrated that Vangl2 deficiency can affect developmental processes. It would be importance to investigate whether Vangl2-dependent NF-κB is influenced by developmental pathways. Distinct from previous studies, our findings revealed no significant differences in spleen and lymph node size between WT and *Vangl2*^ΔM mice. However, myeloid-specific loss of Vangl2 resulted in increased numbers of monocytes, macrophages, and neutrophils in the spleen and BM. This suggests that the ablation of Vangl2 in myeloid cells does not affect the development of immune system development, but rather affects the differentiation of specific cell types. Moreover, we found that Vangl2 is induced and functions under LPS stimulation. Collectively, these findings suggest that Vangl2-dependent NF-κB functions do not affect the development of the immune system, but rather influence the progression of inflammatory-related diseases.

In our study, LPS stimulation did not mediate the expression of Vangl2 in the lungs and liver, though there are lots of resident immune cells in these organs, which suggests that Vangl2 may be important in circulating immune cells and not in the resident population. Given that LPS stimulation markedly alters the expression of Vangl2 in lymph nodes and spleen, rather than in the lungs and liver, we concentrated on the function of Vangl2 in myeloid cells. However, the restricted tissue specificity of the interaction between two ubiquitously present proteins remains a challenge to comprehend and further investigation is needed.

The proteasome, lysosome, and autolysosome pathways are the major systems that are utilized by eukaryotic cells to maintain the protein abundance and immune homeostasis (*Deretic, 2021*). Previous studies have shown that cellular levels of Vangl2 (*Feng et al., 2021*) and its binding partner Prickle2 (*Nagaoka et al., 2019*) are maintained by the proteasomal pathway. Our recent study showed that Vangl2-mediated osteogenic differentiation by limiting CMA in mesenchymal stem cells, suggesting a potential close relationship between Vangl2 and autophagy (*Gong et al., 2021*). Moreover, Vangl2 interacts with LAMP-2A through PkBD domain. Interestingly, we also found Vangl2 interacted with p65 through PkBD domain, indicating that the PkBD domain of Vangl2 play a pivotal role in mediating the interaction between Vangl2 and its target protein. Furthermore, our study suggests Vangl2 promotes p65 degradation through autophagy pathway, but not proteasomal pathway. Importantly, we demonstrated that the degradation of p65 mediated by Vangl2–NDP52 complex is regulated by autophagy induction through rapamycin treatment and autophagy blockade by ATG5 KO or Beclin1 KO. Considering our findings and previous reports, Vangl2 may play multifunctional roles in regulating different types of autophagy (i.e. CMA or selective autophagy).

Selective autophagy requires that cargo receptors recognize the labeling of cargoes with degradation signals and subsequently engaged with the LC3 localized in the autophagosome membrane (*Shaid et al., 2013*). Common cargo receptors mainly include p62/SQSTM1, CALCOCO2/NDP52, OPTN, NBR1, and TOLLIP (*Johansen and Lamark, 2011*; *Yamano and Youle, 2020*). A study in HEK293T cells showed that LRRC25 promotes the autophagic degradation of p65 through enhancing the interaction between p65 and p62 (*Feng et al., 2017*). In addition, a recent study showed that Vangl2 interacts with p62, subsequently promoting breast tumors by activating JNK signaling (*Puvirajesinghe et al., 2016*). Thus, we hypothesized that Vangl2 may promote the autophagic degradation of p65 by recruiting the cargo receptor p62. However, our data suggested that Vangl2 markedly increased the p65–NDP52 interaction but not p65–p62. Strikingly, Vangl2-mediated autophagic degradation of p65 was abolished in NDP52 KO cells, but not in p62 KO cells. Thus, our findings identify that NDP52 is the new cargo receptor responsible for Vangl2-mediated selective autophagic degradation of p65.

Accumulating evidence has shown that cargo receptors mainly recognize ubiquitination modifications on the substrates and then promote degradation in auto-lysosome (*Shaid et al., 2013*). As expected, we observed Vangl2 promoted the ubiquitination of p65, thus enhancing the association between cargo protein and p65. K48- and K63-linked ubiquitination were mostly reported modifications on p65 (*Kauppinen et al., 2013*; *Korbecki et al., 2019*). For example, E3 ubiquitin ligase RNF182 inhibited TLR-triggered cytokine production by promoting K48-linked poly-ubiquitination of p65 (*Cao et al., 2019*). Trim21 promoted K63-linked poly-ubiquitination of p65, but did not affect the stability of p65 (*Yang et al., 2021*). PDLIM7 cooperated with PDLIM2 to inhibit inflammation by promoting K63-linked poly-ubiquitination of p65 (*Jodo et al., 2020*). In this study, we revealed that Vangl2 specifically promoted K63-linked poly-ubiquitination of p65, but not other ubiquitin linkages

by recruiting a previously unrecognized E3 ligase PDLIM2, which further explore the molecular mechanism by which regulates the stability of p65. Although many studies focus on the regulation of p65, the location of p65 degradation has not been clarified. A study showed PDLIM2 degrades p65 through the proteasomal pathway in the nucleus (*Jodo et al., 2020*), while we found Vangl2-mediated autophagic degradation of p65 mainly happened in cytoplasm, but not in nucleus. Combined with previous studies, our study indicates that p65 undergoes different degradation pathway in distinct location of cells.

Autophagy is a fundamental biological process contributing to multiple life processes. Emerging evidence has suggested that crosstalk between autophagy and innate immune signaling plays critical roles in diseases with inflammatory components, including infections, cancer, autoimmunity, and metabolic disorders (*Pradel et al., 2020*; *Wu et al., 2021*). Recent study showed that the interplay between autophagy and type I IFN or NF-κB signaling drives or suppresses inflammatory responses during SARS-CoV-2 infection (*Hui et al., 2021*). Moreover, autophagy-related key genes, such as Atg5, Atg9, and ULK1, also play critical roles in inflammatory diseases (*He et al., 2018*; *Peng et al., 2019*). Here, our discovery of Vangl2–PDLIM2–NDP52 complex in the regulation of p65-mediated NF-κB signaling could be a therapeutic target for the development of immunotherapy against infection and inflammation.

The IKK complex plays a pivotal role in the phosphorylation of p65, which in turn influences NF-κB transcriptional activity. Our findings also indicate that Vangl2 deficiency results in an increased accumulation of phosphorylated p65 and IKK also at 30 min post-LPS stimulation. However, autophagic degradation of p65 may not have been initiated at this early time point. The function of Vangl2 in mediating NF-κB activation during the early stages of LPS stimulation may be complex. Consequently, these data suggest the intriguing possibility that Vangl2 may also regulate the immediate early phase of the inflammatory response via the IKK–p65 axis, which need further study.

Based on our findings, we propose a working model that Vangl2 negatively regulates NF-κB signaling (*Figure 7I*). Vangl2 functions as an adaptor to recruit ubiquitin ligase PDLIM2 and increase K63-linked ubiquitination on p65, which promotes the recognition of p65 by cargo receptor NDP52 and the autophagic degradation of p65, resulting in suppressing the production of proinflammatory cytokines and ameliorating sepsis. Our findings provide a potential target for the treatment of inflammatory diseases.

## Materials and methods
### Animal and sepsis model
All animal experiments were approved by the Southern Medical University Animal Care and Use Committee (SMUL20201010). The *Vangl2*flox/flox mice and lysozyme-Cre (*Lyz2*-Cre) mice were obtained from the Jackson Laboratory (Bar Harbor, ME, USA; Jax no. 025174 and 004781, respectively). *Vangl-2*flox/flox mice were hybridized with *Lyz2*-Cre mice to obtain *Vangl2*flox/flox*Lyz2*-Cre (*Vangl2*ΔM) mice with *Vangl2* specific deficiency in myeloid cells. Co-housed littermate controls with normal *Vangl2* expression were used as WT. For sepsis model, 10-week-old mice were intraperitoneally (i.p.) injected with LPS (25–30 mg/kg), and survival rate of mice was continuously observed. For other detection, mice with sepsis were sacrificed and samples were collected at indicated time points.

### Isolation of immune cells
For BMDMs, BM cells were collected from the femur and tibia, and then maintained in 20% L929 conditioned media with 1% penicillin–streptomycin and macrophage-colony stimulating factor (St Louis, MO, USA) for 6 days, as previously described (*Tan et al., 2019*). Mouse pMACs were acquired from ascites of indicated mice, which were administrated i.p. with 4% (vol/vol) thioglycollate (BD) for three consecutive days before sacrifice. As for primary neutrophils, mice were administrated (i.p.) with 4% (vol/vol) thioglycollate (BD) for 4 hr before sacrifice. Peritoneal cavities were flushed with PBS to obtain pMACs or neutrophils. Cells were cultured in complete Dulbecco's modified Eagle's medium (DMEM) (Corning) supplemented with 100 U/ml penicillin (Sigma), 100 μg/ml streptomycin (Sigma), and 10% fetal bovine serum (FBS) (HyClone) for 6 hr and then washed twice with PBS to remove adherent cells. For human PBMC isolation, 4 ml *Figure 1—source data 1* of whole blood was harvested from each sepsis patient and healthy control. The white membrane layer was collected after

density gradient centrifugation at 400 × g for 25 min and the PBMCs were filtered and prepared for the subsequent experiments after the removal of the red blood cells. For mouse CD11b[+] splenocytes isolation, spleens were digested in 2% FBS–DMEM with 200 U/ml DNase I (Sigma, USA) and 1 mg/ml collagenase type II (Sigma, USA) at 37°C for 30 min. Then tissues digested were filtered through a 70-µm cell strainer and red cells were removed by ACK. The rest splenocytes were labeled with anti-mouse CD11b biotin antibodies (Biolegend). The mixture above was then incubated with streptavidin-paramagnetic particles (BD Biosciences) at 4°C for 30 min. Purification of CD11b[+] splenocytes was performed by DynaMag (Thermo Fisher Science). Isolated cells achieved a purity of ≥95% measured by fluorescence-activated cell sorting (FACS).

## Cell isolation and culture

All cells were obtained from and authenticated by ATCC and determined to be mycoplasma free. HEK293T (#CRL-11268), CHO (#CCL-61), and A594 (#CCL-185) cells were cultured according to ATCC guidelines at low passage number (less than 10 passages; typically passages 2–4). Above cell lines were maintained in complete DMEM (Bio-Channel), supplemented with 10% FBS (Vazyme, Nanjing, China), 100 U/ml penicillin and streptomycin. All cells were cultured at 37°C with 5% $CO_2$.

## Cell treatment

To test cytokines expression and signaling pathway activation, BMDMs, neutrophils and A549 cells were treated with LPS (200 ng/ml) for the indicated time. For dual luciferase assay, HEK293T cells were treated with LPS (250 ng/ml), IL-1β (40 ng/ml), or TNF-α (20 ng/ml) for 6 hr. For protein degradation inhibition assays in HEK293T cells, CQ (50 µM), 3-MA (10 mM), or bafilomycin A1 (Baf A1) (0.2 µM) (AmBeed) was used to inhibit autolysosome- or lysosome-mediated protein degradation. MG132 (10 µM) (AmBeed) was used to inhibit proteasome-mediated protein degradation. Z-VAD-FMK (50 µM) (AmBeed) was used to inhibit caspase-mediated protein degradation.

## Luciferase and reporter assays

HEK293T cells were plated in 24-well plates and transfected with pRL-TK and plasmids encoding the NF-κB luciferase reporter, together with different plasmids following: Flag-MyD88, Flag-IRAK1, Flag-TRAF6, Flag-IKKα, Flag-IKKβ, Flag-p65, and an increasing dose of the HA-Vangl2 vector (250, 500, and 1000 ng) or empty vector. In addition, CHO or HEK293T cells were administrated with or without LPS (1000 ng/ml), IL-1β (1000 ng/ml), or TNF-α (100 ng/ml) for 6–8 hr after transfection with pRL-TK and plasmids encoding the NF-κB luciferase reporter. Then, cells were collected at 24 hr post-transfection and luciferase activity was analyzed by Dual-Luciferase Reporter Assay Kit (Vazyme) performed with a Luminoskan Ascent luminometer (Thermo Fisher Scientific). The activity of Firefly luciferase was normalized by that of Renilla luciferase to obtain relative luciferase activity.

## Enzyme-linked immunosorbent assay

IL-1β, IL-6, and TNF-α in cell supernatants and mice serum were measured using ELISA kit (#E-EL-M0037c, #E-EL-M0044c, and #E-EL-M1084c, respectively; Elabscience Biotechnology) following the manufacturer's instructions. Absorbance was detected at 450 nm by the Multiscan FC (Thermo Fisher, Waltham, MA, USA).

## Immunoprecipitation and immunoblot analyses

Cells were lysed by low-salt lysis buffer (NCM Biotech). For endogenous immunoprecipitation, whole-cell lysates were treated with indicated antibodies overnight and then incubated protein A/G beads (Pierce) for 4–6 hr. For exogenous immunoprecipitation, whole-cell lysates were incubated with anti-FLAG or anti-Myc agarose gels. Immunoprecipitates were eluted with 2× sodium dodecyl sulfate (SDS) loading buffer after five times washing with low-salt lysis buffer. The proteins were dissolved in SDS loading buffer and boiled for 8–10 min. Then protein lysates resolved on SDS–polyacrylamide gel electrophoresis (PAGE) gels and proteins were transferred to a polyvinylidene difluoride membrane (Millipore). After blocking with 5% (wt/vol) reagent-grade nonfat milk (Sigma), the membranes were incubated with the indicated antibodies (*Supplementary file 1*) with universal antibody diluent (NCM Biotech) overnight. For all blots, proteins were detected by EMD Millipore Luminata Western HRP Chemiluminescence Substrate.

## Flow cytometry analysis

Mouse splenocytes were stained with indicated antibodies (*Supplementary file 1*) at 4°C in roswell park memorial institute (RPMI) containing 2% FBS for 30–60 min. All samples were detected by BD LSRFortessa flow cytometry analyzer (BD sciences). The data were analyzed via FlowJo X software (Tree Star).

## RNA preparation and qPCR

Total RNA was purified from stimulated cells and splenic tissue by the TRIzol reagent (Invitrogen), and cDNA was obtained using starscript II first-stand cDNA synthesis kit (GenStar, Beijing, China). Real-time PCR was performed on QuantStudio 6 flex (Thermo Fisher, Waltham, MA, USA) using RealStar green power mixture (GenStar, Beijing, China) with primers listed in *Supplementary file 2*.

## siRNA transfection

*Pdlim2*, *Usp7*, and *Trim21* siRNA were obtained from Genechem (Shanghai, China) (*Supplementary file 3*). Cells were transfected with siRNA or scrambled shRNA at 2 µg per well for 24-well plates (Promth). Transfection of HEK293T and BMDM were performed using Lipofectamine 2000 (Invitrogen) according to the manufacturer's recommended protocols.

## Sepsis patients' information

Whole blood samples from patients who were diagnosed as sepsis induced by gram-negative bacterial infection were obtained from Nanfang Hospital, Southern Medical University. Criteria for enrolled patients followed by the international guidelines for management of sepsis and septic shock of 2016. The Sepsis patients' information is shown in *Supplementary file 4*. Before the collection of whole blood samples, the informed consents of the patients were obtained. These consents included the voluntary donation of blood tissue, and consent for the use of all specimens for scientific research and for publication of the results obtained in scientific journals. This project was implemented by the approval of the Ethics Committee of Nanfang Hospital, Southern Medical University (registered number NFEC-2023-437).

## Cellular fractionation

Cells were collected by scraping, spun down and washed in pre-chilled PBS. For cytoplasmic and nuclear extracts were prepared by NE-PER Nuclear and Cytoplasmic Extraction Reagents (Thermo). Briefly, cytoplasmic were extracted by ice-cold CER I and CER II reagent, and nuclear were extracted by ice-cold CER I and CER II NER reagent. For cytosol and membrane, MELB buffer (20 mM 4-(2-hydroxyethyl) piperazine-1-ethanesulfonic acid pH = 7.5, 100 mM sucrose, 2.5 mM $MgCl_2$, 100 mM KCl) containing 0.025% digitonin was used to permeabilize cells to extract the cytosol fraction, and 1% digitonin buffer was used to extract cell membrane fraction.

## Statistical analysis

The data of all quantitative experiments are presented as mean ± standard deviation of at least three independent experiments. Curve data were assessed by GraphPad Prism 8.0 (USA). And comparisons between groups for statistically significant differences were analyzed with a two-tailed Student's *t* test. The statistical significance was defined as $p < 0.05$.

## Online supplemental material

*Figure 1—figure supplement 1* shows that the expression of Vangl2 during sepsis and LPS treatment.

*Figure 2—figure supplement 1* shows that Vangl2 defection promotes LPS-induced NF-κB activation and production of inflammatory cytokines.

*Figure 3—figure supplement 1* shows that Vangl2 interacts with p65 to inhibit NF-κB activation.

*Figure 4—figure supplement 1* shows that Vangl2 promotes p65 degradation by autophagic pathway.

*Figure 5—figure supplement 1* shows that Vangl2 promotes autophagic degradation of p65 is not mediated by cargo receptor p62.

*Figure 7—figure supplement 1* shows that expression of candidate E3 ubiquitin ligases in WT and Vangl2-deficient BMDMs after LPS treatment.

Supplementary file 1 shows reagents and antibodies used in this study.
Supplementary file 2 shows primers sequences for quantitative RT-PCR.
Supplementary file 3 shows primers sequences for siRNA transfection.
Supplementary file 4 shows the sepsis patients' information.

## Acknowledgements

We thank Dr. Jun Cui (Sun Yat-sen University) for providing ATG5, Beclin1, NDP52, and p62 KO HEK293T cells. We thank Experimental Education/Administration Center of School of Basic Medical Science of Southern Medical University for providing assistance. This work was supported by grants from National Natural Science Foundation of China (82171741, 82371761, and 82302731), Guangdong Basic and Applied Basic Research Foundation (2021A1515012140), and China Postdoctoral Science Foundation (2021M701622).

## Additional information

### Funding

| Funder | Grant reference number | Author |
|---|---|---|
| National Natural Science Foundation of China | 82302731 | Jiansen Lu |
| China Postdoctoral Science Foundation | 2021M701622 | Jiansen Lu |
| National Natural Science Foundation of China | 82171741 | Xiao Yu |
| National Natural Science Foundation of China | 82371761 | Xiao Yu |
| Guangdong Basic and Applied Basic Research Foundation | 2021A1515012140 | Xiao Yu |

The funders had no role in study design, data collection, and interpretation, or the decision to submit the work for publication.

### Author contributions

Jiansen Lu, Data curation, Funding acquisition, Writing – original draft, Writing – review and editing; Jiahuan Zhang, Conceptualization, Formal analysis, Methodology, Writing – original draft; Huaji Jiang, Resources, Software, Writing – review and editing; Zhiqiang Hu, Conceptualization, Formal analysis, Methodology, Writing – review and editing; Yufen Zhang, Validation, Visualization; Lian He, Software, Methodology; Jianwu Yang, Qingyue Xiao, Resources; Yingchao Xie, Validation; Dan Wu, Hongyu Li, Investigation; Ke Zeng, Formal analysis; Peng Tan, Conceptualization; Zijing Song, Chenglong Pan, Software; Xiaochun Bai, Supervision, Writing – original draft, Project administration, Writing – review and editing; Xiao Yu, Supervision, Funding acquisition, Writing – original draft, Project administration, Writing – review and editing

### Author ORCIDs

Jiansen Lu (iD) https://orcid.org/0000-0003-3875-5583
Hongyu Li (iD) https://orcid.org/0000-0003-1784-4595
Xiaochun Bai (iD) https://orcid.org/0000-0001-9631-4781
Xiao Yu (iD) https://orcid.org/0000-0003-2491-9110

### Ethics

Human blood were obtained from Sepsis and Health checkup, who had signed informed consents (NFEC-2023-437).
All animal experiments were approved by the Southern Medical University Animal Care and Use Committee (SMUL20201010).

Reviewer #1 (Public Review): https://doi.org/10.7554/eLife.87935.4.sa1
Reviewer #2 (Public Review): https://doi.org/10.7554/eLife.87935.4.sa2
Reviewer #3 (Public Review): https://doi.org/10.7554/eLife.87935.4.sa3
Author response https://doi.org/10.7554/eLife.87935.4.sa4

## Additional files

### Supplementary files
• Supplementary file 1. Reagents and antibodies used in this study.

• Supplementary file 2. Primers sequences for quantitative real-time reverse transcription polymerase chain reaction (RT-PCR) .

• Supplementary file 3. Primers sequences for siRNA transfection.

• Supplementary file 4. The sepsis patients' information was shown in this study.

• MDAR checklist

### Data availability
All original gel showed in supplemental data. *Figure 2—source data 1*, *Figure 3—source data 1*, and *Figure 4—figure supplement 1—source data 1* contain the numerical data used to generate the figures.

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
