## [Editor Report · eLife assessment]

This **valuable** manuscript describes a novel role of Vangl2, a core planar cell polarity protein, in linking the NF-kB pathway to selective autophagic protein degradation in myeloid cells. The mechanistic studies provide **convincing** evidence that Vangl2 targets p65 for NDP52-mediated autophagic degradation, limiting inflammatory NF-kB response, with functional significance of the proposed mechanism in sepsis. Additional future studies dissecting autophagic Vangl2 functions in various myeloid subsets in the context of inflammation could be informative, and additional Vangl2 targets in the inflammatory pathway, including IKK2, could also be explored. Overall, this exciting study can advance our understanding of NF-kB control, particularly in the context of inflammatory diseases.

---

## [Referee Report · Reviewer #1 (Public Review)]

The study shows a new mechanism of NFkB-p65 regulation mediated by Vangl2-dependent autophagic targeting. Autophagic regulation of p65 has been reported earlier; this study brings an additional set of molecular players involved in this important regulatory event, which may have implications for chronic and acute inflammatory conditions.

---

## [Referee Report · Reviewer #2 (Public Review)]

Vangl2, a core planar cell polarity protein involved in Wnt/PCP signaling, cell proliferation, differentiation, homeostasis, and cell migration. Vangl2 malfunctioning has been linked to various human ailments, including autoimmune and neoplastic disorders. Interestingly, it was shown that Vangl2 interacts with the autophagy regulator p62, and autophagic degradation limits the activity of inflammatory mediators, such as p65/NF-κB. However, the possible role of Vangl2 in inflammation has not been investigated. In this manuscript, Lu et al. describe that Vangl2 expression is upregulated in human sepsis-associated PBMCs and that Vangl2 mitigates experimental sepsis in mice by negatively regulating p65/NF-κB signaling in myeloid cells. Their mechanistic studies further revealed that Vangl2 recruits the E3 ubiquitin ligase PDLIM2 to promote K63-linked poly-ubiquitination of p65. Vangl2 also facilitated the recognition of ubiquitinated p65 by the cargo receptor NDP52. These molecular processes caused selective autophagic degradation of p65. Indeed, abrogation of PDLIM2 or NDP52 functions rescued p65 from autophagic degradation, leading to extended p65/NF-κB activity in myeloid cells. Overall, the manuscript presents convincing evidence for novel Vangl2-mediated control of inflammatory p65/NF-kB activity. The proposed pathway may expand interventional opportunities restraining aberrant p65/NF-kB activity in human ailments.

IKK is known to mediate p65 phosphorylation, which instructs NF-kB transcriptional activity. In this manuscript, Vangl2 deficiency led to an increased accumulation of phosphorylated p65 and IKK also at 30 minutes post-LPS stimulation; however, autophagic degradation of p-p65 may not have been initiated at this early time point. Therefore, this set of data put forward the exciting possibility that Vangl2 could also be regulating the immediate early phase of inflammatory response involving the IKK-p65 axis - a proposition that may be tested in future studies.

---

## [Referee Report · Reviewer #3 (Public Review)]

Lu et al. describe Vangl2 as a negative regulator of inflammation in myeloid cells. The primary mechanism appears to be through binding p65 and promoting its degradation, albeit in an unusual autolysosome/autophagy dependent manner. Overall, these findings are novel, valuable and the crosstalk of PCP pathway protein Vangl2 with NF-kappaB is of interest.

Comments on latest version:

Lu et al. now address all my comments. All data included for the reviewers should be included in the main manuscript or Supplement and should be available to the readers. Please ensure that this criteria is met. I have no further comments.

---

## [Author Response]

The following is the authors’ response to the previous reviews.

**Responses to Reviewer #1:**
Reviewer #1: The study shows a new mechanism of NFkB-p65 regulation mediated by Vangl2-dependent autophagic targeting. Autophagic regulation of p65 has been reported earlier; this study brings an additional set of molecular players involved in this important regulatory event, which may have implications for chronic and acute inflammatory conditions.Comments on the revised version:The authors have addressed the earlier concerns and I am satisfied with the revised version. I have no additional comments to make.

We appreciate the reviewer’s comments on our revised manuscript.

**Responses to Reviewer #2:**
Reviewer #2: Vangl2, a core planar cell polarity protein involved in Wnt/PCP signaling, cell proliferation, differentiation, homeostasis, and cell migration. Vangl2 malfunctioning has been linked to various human ailments, including autoimmune and neoplastic disorders. Interestingly, it was shown that Vangl2 interacts with the autophagy regulator p62, and autophagic degradation limits the activity of inflammatory mediators, such as p65/NF-κB. However, the possible role of Vangl2 in inflammation has not been investigated. In this manuscript, Lu et al. describe that Vangl2 expression is upregulated in human sepsis-associated PBMCs and that Vangl2 mitigates experimental sepsis in mice by negatively regulating p65/NF-κB signaling in myeloid cells. Their mechanistic studies further revealed that Vangl2 recruits the E3 ubiquitin ligase PDLIM2 to promote K63-linked poly-ubiquitination of p65. Vangl2 also facilitated the recognition of ubiquitinated p65 by the cargo receptor NDP52. These molecular processes caused selective autophagic degradation of p65. Indeed, abrogation of PDLIM2 or NDP52 functions rescued p65 from autophagic degradation, leading to extended p65/NF-κB activity in myeloid cells. Overall, the manuscript presents convincing evidence for novel Vangl2-mediated control of inflammatory p65/NF-kB activity. The proposed pathway may expand interventional opportunities restraining aberrant p65/NF-kB activity in human ailments.IKK is known to mediate p65 phosphorylation, which instructs NF-kB transcriptional activity. In this manuscript, Vangl2 deficiency led to an increased accumulation of phosphorylated p65 and IKK also at 30 minutes post-LPS stimulation; however, autophagic degradation of p-p65 may not have been initiated at this early time point. Therefore, this set of data put forward the exciting possibility that Vangl2 could also be regulating the immediate early phase of inflammatory response involving the IKK-p65 axis - a proposition that may be tested in future studies.

We appreciate the reviewer’s comments on our manuscript, and we have added the discussion about IKK-p65 axis in revised version. (Page 15, lines 467-474)

**Responses to Reviewer #3:**
Reviewer #3: Lu et al. describe Vangl2 as a negative regulator of inflammation in myeloid cells. The primary mechanism appears to be through binding p65 and promoting its degradation, albeit in an unusual autolysosome/autophagy dependent manner. Overall, these findings are novel, valuable and the crosstalk of PCP pathway protein Vangl2 with NF-kappaB is of interest. While generally solid, some concerns still remain about the rigor and conclusions drawn.Comments on the revised version:(1) Lu et al. address my comments through responses and new experimental data. However, some of the explanations provided are inadequate.However, in response to my enquiry regarding directly exploring PCP effects, the authors simply assert "Our study revealed that Vangl2 recruits the E3 ubiquitin ligase PDLIM2 to facilitate K63-linked ubiquitination of p65, which is subsequently recognized by autophagy receptor NDP52 and then promotes the autophagic degradation of p65. Our findings by using autophagy inhibitors and autophagic-deficient cells indicate that Vangl2 regulates NFkB signaling through a selective autophagic pathway, rather than affecting the PCP pathway, WNT, HH/GLI, Fat-Dachsous or even mechanical tension."I do not agree that the use of autophagy inhibitors and autophagy-deficient cells can rule out the contributions of PCP or any other pathways. Only experimentally inhibiting the pathway(s) with adequate demonstration of target inhibition/abolition of well-known effector function and documenting unaltered p65 regulation under these conditions can be considered proof. Autophagy inhibitors and autophagy-deficient cells only prove that this particular pathway is necessary. Nonetheless, I do not want to dwell on proving a negative and agree that Vangl2 is a novel regulator of p65 through its role in promoting p65 degradation. The inclusion of a statement discussing the limitations of their approach would have sufficed. The response from the authors could have been better.

We thank the reviewer for helping us improve the quality of the manuscript. We provided new data and revised the Discussion as suggested.

To ascertain whether Vangl2 degrades p65 through a selective autophagic pathway or the PCP pathway, 293T cells were transfected with p65, together with or without the Vangl2 plasmids, and treated with different pharmacological inhibitors. We found the degradation of p65 induced by Vangl2 was blocked by autolysosome inhibitor (CQ), but not by the JNK inhibitor (SP600125) or Wnt/β-catenin inhibitor (FH535) (New Figure. 1). These data suggest that Vangl2 primarily degrades p65 through a selective autophagic pathway, rather than through the JNK or Wnt signaling pathway. Nevertheless, additional pathway inhibitions, such as those of the HH/GLI and Fat-Dachsous pathways, should also be employed to further elucidate the function of Vangl2 in p65 degradation. As suggested, we have added a statement about the limitation of the approach in the discussion (Page 12, lines 378-385).

**Author response image 1. sa4fig1:** Vangl2 degrades p65 through a selective autophagic pathway, but not by the PCP pathway. HEK293T cells were transfected with Flag-p65 and HA-Vangl2 plasmids, and treated with DMSO, CQ (50 mM) for 6 h, SP600125 (20 mM) for 1 h or FH535 (30 mM) for 6 h. The cell lysates were analyzed by immunoblot.

(2) I am also not satisfied with the explanation that "immune cells represent a minor fraction of the lungs and liver". There are lots of resident immune cells in the lungs and liver (alveolar macrophages in the lung and Kuppfer cells in the liver). For example, it may be so that Vangl2 is important in monocytes and not in the resident population. This might be a potential explanation. But this is not explored. The restricted tissue-specificity of the interaction between two ubiquitously present proteins is still a challenge to understand. The response from the authors is not satisfactory. There is plenty of Vangl2 in the liver in their western blot.

We thank the reviewer for this question. We added this explanation in the Discussion. (Page 13, lines 398-404)

(3) I had also simply pointed out PMID: 34214490 with reference to the findings described in the manuscript. There were no suggestions of contradiction. In fact, I would refer to the publication in discussion to support the findings and stress the novelty. The response from the authors could have been better.

Thank you for the reviewer's insightful comments. We have modified this discussion as suggested. (Page 13, lines 410-415; Page 14, lines 419-421)

(4) The response to my enquiry regarding homo- or heterozygosity is unsupported by any reference or data.

As suggested, we provided the data that only Vangl2 deficient homozygous showed inhibition of the activation of NF-kB in New Figure. 2.

**Author response image 2. sa4fig2:** Vangl2 deficiency promotes NF-kB activation. (A) The survival rates of WT, *Vangl2ΔM/ΔM* and *Vangl2ΔM/WT* mice treated with high-dosage of LPS (30 mg/kg, i.p.) (n≥4). (B) IL-6 and TNF-a secretion by WT and Vangl2-deficient BMDMs treated with LPS for 6 h was measured by ELISA. IL-1β secretion by WT, *Vangl2ΔM/ΔM* and *Vangl2ΔM/WT* BMDMs treated with LPS for 6 h and ATP for 30 min was measured by ELISA.

(5) The listing of 8 patients and healthy controls are also appreciated. The body temperature of #6 doesn't fall in the <36 or >38 degree C SIRS criteria. The inclusion of CRP, PCT, heart rate and respiratory rate, and other lab values would have further improved the inclusion criteria. Moreover, it is difficult to understand why there are 16 value points for healthy and sepsis cohorts in Fig 1 when there are 8 patients.

We thank the reviewer for this valuable suggestion. We are sorry for our mistake that we entered data from two repeated experiments in Figure. 1 A and we have revised this data in the updated version (Figure. 1 A, Pages 12 Lines 146). As suggested, we have added CRP, WBC and heart rate in sepsis patients’ information. (Supplementary Materials and Methods)

**Recommendations for the authors:**

**Reviewer #2 (Recommendations For The Authors):**
The proposition that Vangl2 may target additional mediators of inflammation could be indicated in the text.

We thank the reviewer for this valuable suggestion. We had added discussion in modified version. (Page 15, lines 467-474)

**Reviewer #3 (Recommendations For The Authors):**
It is advised that some of the deficiencies pointed out by Reviewer #3 are textually addressed. Additionally, there could be some inconsistency in the number of healthy controls and patients (see Fig S1A and FIg 1A and Supplementary table, also see comments from Reviewer #3) - this should be carefully scrutinised and revised, if necessary.

We thank the reviewer for this valuable suggestion. We are sorry for our mistake that we entered data from two repeated experiments in Figure. 1 A and we have revised this data in the updated version (Figure. 1 A, Pages 12 Lines 146).